# Doubly Robust Augmented Transfer for Meta-Reinforcement Learning

**Yuankun Jiang**[1], **Nuowen Kan**[1], **Chenglin Li**[2], **Wenrui Dai**[1], **Junni Zou**[1], **Hongkai Xiong**[2]
[1]Department of Computer Science and Engineering, [2]Department of Electronic Engineering[*]
Shanghai Jiao Tong University

## Abstract

Meta-reinforcement learning (Meta-RL), though enabling a fast adaptation to learn new skills by exploiting the common structure shared among different tasks, suffers performance degradation in the sparse-reward setting. Current hindsight-based sample transfer approaches can alleviate this issue by transferring relabeled trajectories from other tasks to a new task so as to provide informative experience for the target reward function, but are unfortunately constrained with the unrealistic assumption that tasks differ only in reward functions. In this paper, we propose a doubly robust augmented transfer (DRaT) approach, aiming at addressing the more general sparse reward meta-RL scenario with both dynamics mismatches and varying reward functions across tasks. Specifically, we design a doubly robust augmented estimator for efficient value-function evaluation, which tackles dynamics mismatches with the optimal importance weight of transition distributions achieved by minimizing the theoretically derived upper bound of mean squared error (MSE) between the estimated values of transferred samples and their true values in the target task. Due to its intractability, we then propose an interval-based approximation to this optimal importance weight, which is guaranteed to cover the optimum with a constrained and sample-independent upper bound on the MSE approximation error. Based on our theoretical findings, we finally develop a DRaT algorithm for transferring informative samples across tasks during the training of meta-RL. We implement DRaT on an off-policy meta-RL baseline, and empirically show that it significantly outperforms other hindsight-based approaches on various sparse-reward MuJoCo locomotion tasks with varying dynamics and reward functions.

## 1 Introduction

Reinforcement learning (RL) has achieved a remarkable success in a variety of sequential decision making tasks, such as intelligent gaming [1] and robotic control [2]. Agents trained by conventional RL methods aim at learning a single task, thus fail to adapt quickly to a new task with prior experience. In contrast, meta-reinforcement learning (meta-RL) focuses on *learning to learn*, i.e., learning how to adapt [3, 4]. In essence, it learns the underlying common structure from experience collected across a set of tasks, and then exploits this structure to fast adapt to similar new tasks with only a few trials.

Currently, one of the main challenges in meta-RL is the sparse-reward setting, which exists widely in real-world environments. When rewards become sparse, the agent receives only scarce information related to the task, bringing extreme difficulty to the meta-training and adaptation processes. To collect more useful experience, sample transfer approaches have been proposed. The key idea behind is to learn a new task by selectively leveraging samples generated from other tasks. Ideally, samples transferred from an original task to the new target task are required to be informative trajectories.

---

[*]Correspondence to: Chenglin Li <LCL1985@sjtu.edu.cn>, Wenrui Dai <daiwenrui@sjtu.edu.cn>, and Junni Zou <zoujunni@sjtu.edu.cn>.

37th Conference on Neural Information Processing Systems (NeurIPS 2023).

Namely, the informative trajectories should present a higher and denser reward in the target task, though they might be unsuccessful attempts (i.e., with lower and sparser reward) in the original task.

Following the direction of sample transfer, hindsight experience replay (HER) and its variants [5, 6] have been developed as the typical methods in literature. Hindsight-based methods relabel a trajectory collected from the original task to a target task as informative trajectory, where rewards of this trajectory are re-computed using the target reward function to teach it how to achieve a different goal in the new task. However, HER algorithms are inherently designed for tasks that only differ in the reward function, while dynamics of all the different tasks are assumed to be identical. It is worth mentioning that in many real-world scenarios, tasks may differ not only in reward functions, but also in dynamics. For example, a robot is often trained under various working conditions, such as different load mass and joint damping, which may result in different distributions of dynamics. We are thus motivated to consider a more general meta-RL scenario where both dynamics and reward function can be different and varying across tasks, and study how to further cope with such a mismatch of dynamics distributions. This brings new challenges to existing meta-RL baselines, since consideration of only dynamics difference with the identical dense-reward setting fails to handle multi-goal tasks, while considering only the reward difference may misjudge values of transferred trajectories.

In off-policy RL, doubly robust (DR) estimators [7, 8] have emerged as an effective method to correct the distribution mismatch between a target policy and a behavior policy. They incorporate importance sampling with a fitted model of value function to alleviate the high variance issue of the original importance sampling-based estimator. In meta-RL, however, the discrepancy between the dynamics of the original and target tasks (as represented by the importance weight of dynamics) is practically inaccessible to the agent. This indicates that popular methods, such as directly clipping or removing large importance weight values [9, 10], proposed for providing a more accurate evaluation to enable efficient sample transfer and thus stable training, become infeasible in meta-RL.

In order to control the values of dynamics importance weights, we formulate the sample transfer as a mean square error (MSE) minimization problem and solve for the optimal dynamics importance weight. Since this optimal weight is intractable in practice, we propose an approximation strategy that constrains the estimated weight within an interval containing this optimal weight, and theoretically show that any estimate outside this interval would inevitably increase the MSE. Finally, we present a policy learning algorithm for sample transfer-based meta-RL, in which the proposed weight estimate is used for value-function evaluation. Our main contributions can be summarized as follows.

- **Doubly Robust augmented Estimator**. We design a doubly robust augmented estimator (DRaE), which for the first time accommodates to the more general and rational meta-RL setting and simultaneously allows varying dynamics and reward functions across different tasks. DRaE tackles the mismatch of dynamics distributions in meta-RL with a guaranteed optimum for the dynamics importance weight by minimizing MSE between the estimated and true values of the value function.

- **Theoretically Robust Interval Approximation**. We propose a tractable interval-based approximation for the optimal dynamics importance weight derived by our DRaE. This interval is deterministic to balance the variance and bias for a decreasing MSE and guaranteed to cover the optimum. We further verify that the proposed approximation is robust in theory with a constrained and sample-independent upper bound on the MSE approximation error from the optimum.

- **Doubly Robust augmented Transfer Algorithm**. We develop a doubly robust augmented transfer (DRaT) algorithm for transferring informative samples across tasks, based on the proposed DRaE. DRaT controls the importance weight under the guide of our interval-based approximation to minimize the MSE. It is noted that DRaT can be integrated into any value-based meta-RL baselines and implemented specifically on an off-policy meta-RL baseline, PEARL [4].

## 2 Preliminary

### 2.1 Meta-Reinforcement Learning (Meta-RL)

Meta-RL considers training on a task set $\{\mathcal{T}_i\}$ to learn a set of meta-parameters, which can quickly adapt to solving a new (testing) task with only a limited number of samples. Each task $\mathcal{T}_i$ then implies a Markov decision process (MDP) with both a distinct reward function $r_i$ and a different transition probability distribution $p_i(\cdot|s, a)$. Such a meta-RL objective can be written as: $\max_\theta \mathbb{E}_{P_\mathcal{T}} \left[ J_\mathcal{T}(\pi_{\theta_{\mathcal{T}_i}}) \right]$, where $P_\mathcal{T}$ denotes the sampling distribution over the training task set $\mathcal{T}$, $J_\mathcal{T}(\pi_{\theta_{\mathcal{T}_i}})$ is the expected

reward of policy$\pi_{\theta_{\mathcal{T}_i}}$ on the task $\mathcal{T}_i$, and the policy $\pi_{\theta_{\mathcal{T}_i}}$ is generated from $\pi_\theta$ following a certain adaptation procedure given samples $c_i = \{s_t, a_t, r_t, s_{t+1}\}$ from this task.

We use PEARL [4] specifically in this paper as the baseline meta-RL algorithm, which builds upon the soft actor-critic (SAC) [11]. Nevertheless, our proposed DRaT approach is compatible with and can be applied generally to any other meta-RL baselines. PEARL implements the adaptation by constructing an inference network $q_\phi(z|c)$, which takes in recently collected samples $c$ of a task (also referred to as the context) and infers a latent context variable $z$ that encodes salient information about this task. Then, a policy $\pi_\theta(a|s, z)$ conditioned on $z$ can adapt its behavior to this task. Both the inference network $q_\phi(z|c)$ and critic $Q_\theta(s, a, z)$ are trained with the following critic loss:

$$\mathcal{L}_{critic} = \frac{1}{2}\mathbb{E}_{(s,a,r,s')\sim\mathcal{B}, z\sim q_\phi(z|c)}\left[Q_\theta(s,a,z) - (r(s,a) + \gamma\bar{V}_\theta(s',\bar{z}))\right]^2, \tag{1}$$

where $\bar{V}_\theta$ is the target value network and the overline notation specifies no gradient backpropagation going through $\bar{V}_\theta$ and $\bar{z}$. While the policy is trained using the same actor loss as in SAC [11].

## 2.2 Doubly Robust Estimator for Off-Policy Policy Evaluation

In off-policy policy evaluation, we expect to estimate the state value of a target policy $\pi$ given data sampled from a behavior policy $\mu$, where distribution mismatch stems from varying polices. Importance sampling (IS) estimators can correct this mismatch by multiplying the importance weights of policies, thus providing an unbiased estimate: $V^{IS}(s_t = s) = \left[\prod_{k=t}^{T-1}\rho_\pi(k)\right] \cdot \left[\sum_{k=0}^{T-t-1}\gamma^k r(s_{t+k}, a_{t+k})\right]$, where $\rho_\pi(k) = \frac{\pi(a_k|s_k)}{\mu(a_k|s_k)}$ denotes the importance weight between the target policy $\pi$ and behavior policy $\mu$ at the step $k$. Unfortunately, the IS estimator suffers from high variance due to possible high values of the policy importance weights, while the product of policy importance weights will also grow exponentially with the horizon [7]. Through incorporating a fitted value function into the IS estimator, this variance can be reduced by doubly robust (DR) estimators, which approximate the value-function at $s_t$ recursively as:

$$V^{DR}(s_t = s) = \hat{V}(s) + \rho_\pi(t)\left[r(s_t, a_t) + \gamma V^{DR}(s_{t+1}) - \hat{Q}(s, a_t)\right], \tag{2}$$

where $\hat{V}(s) = \mathbb{E}_{a\sim\pi(\cdot|s)}\left[\hat{Q}(s, a)\right]$ is computed by the fitted $Q$-function, i.e., $\hat{Q}$. Here, "doubly robust" refers to that the estimated value is unbiased when either one of the $\rho_\pi$ and $\hat{Q}$ is an unbiased estimate.

# 3 Doubly Robust Augmented Estimator (DRaE) for Sample Transfer

In this section, we consider leveraging the sample transfer between tasks in a more general sparse-reward meta-RL setting, where both the reward functions and dynamics are varying across different tasks. Specifically, we formulate the sample transfer as a mean square error (MSE) minimization problem between the estimate value of samples transferred from other tasks and their true values in the target task. Other than the simple transfer of relabeled samples in hindsight-based methods without accommodating to the dynamics mismatch, solution to the proposed sample transfer problem can provide the true value estimate of transferred samples for the subsequent tasks in meta-training, which implicitly incorporates the differences of both reward functions and dynamics models.

We then show that a direct use of DR will bring additionally high variance incurred by environment dynamics, since the importance weight of dynamics might be large due to various dynamics. Though the unbiasedness of DR estimate is appealing, its high variance will cause a large MSE on the estimated value, which significantly disturbs the meta-training process. Since MSE can be decomposed into the sum of bias and variance-related terms, we thus propose to minimize a derived upper bound of MSE and determine the optimal dynamics importance weight, aiming at balancing the bias and variance. Finally, to overcome the intractability of solving this MSE minimization in practice, we propose instead an interval-based approximation for the optimal importance weight, by showing that any other values outside this interval will enlarge both the bias and variance, thus with a larger MSE.

## 3.1 Problem Formulation of Sample Transfer in Meta-RL

In meta-RL, a training task set $\mathcal{T}$ can be characterized by tuples $\{\mathcal{T}_i = \langle\mathcal{S}, \mathcal{A}, p_i, r_i\rangle, i = 1, \cdots, N\}$, where $\mathcal{S}$ and $\mathcal{A}$ denote the common state and action spaces, respectively, while a task $\mathcal{T}_i$ is distinct

in both the transition distribution $p_i$ and reward function $r_i$. At each training iteration, informative trajectories that are sampled from other tasks but expected to achieve a higher sum of rewards for a certain task $\mathcal{T}_j$ will be transferred to and leveraged by $\mathcal{T}_j$, which is the core idea behind hindsight-based methods. Here, we argue that their intuitive sum of rewards, however, cannot accurately indicate the true state value $V_j$ of a trajectory that is sampled from other tasks and transferred to this target task, especially when dynamics mismatches arise across tasks. For a more accurate estimation, we therefore formulate the following sample transfer problem by minimizing the MSE between the true state value $V_j$ and its estimate $\hat{V}_j$:

$$\min_{\hat{V}_j} \sum_{\tau_i \sim \mathcal{D}_j} \sum_t \mathbb{E}_{\tau_i|_{t:T}} \left[ \left( V_j(s_t) - \hat{V}_j(s_t) \right)^2 \big| s_t = s \right], \tag{3}$$

where $\mathcal{D}_j = \{ \tau_i = \{s_t, a_t, r_t, s_{t+1}\} \big|_{t=0}^{T} \}_{i \neq j}$ is the dataset that contains informative trajectories sampled from other tasks and selected using any relabeling strategy $\mathcal{S}_I$, $s_t = s$ is the state at step $t$ from a trajectory $\tau_i$, and the expectation is taken w.r.t. the randomness of trajectory $\tau_i$ from step $t$ to the final step $T$ sampled in the task $\mathcal{T}_i$. Further, the MSE objective in Eq. (3) can be decomposed as:

$$\mathbb{E}_{\tau_i|_{t:T}} \left[ \left( V_j(s_t) - \hat{V}_j(s_t) \right)^2 | s_t \right] = \left( V_j(s_t) - \mathbb{E}_{\tau_i|_{t:T}} [\hat{V}_j(s_t)|s_t] \right)^2 + Var \left( \hat{V}_j(s_t) \right), \tag{4}$$

where the first term on the RHS equals square of the bias of $\hat{V}_j$ that is estimated from computation on the informative trajectories, while the second term denotes the variance of estimate $\hat{V}_j$.

## 3.2 Direct Use of Doubly Robust (DR) Estimator

DR estimators can provide an unbiased estimator for off-policy policy evaluation with existence of the distribution mismatch. For a trajectory that is sampled from task $\mathcal{T}_i$ and transferred to a target task $\mathcal{T}_j$, the unbiased estimate for the state value can be determined by directly using the idea of DR:

$$V_{ij}^{DR}(s_t = s) = V_\theta(s, z_j) + \rho_\pi^{ij}(t) \left[ r_j(s, a_t) + \rho_d^{ij}(t+1)\gamma V_{ij}^{DR}(s_{t+1}) - Q_\theta(s, a_t, z_j) \right], \tag{5}$$

where $\rho_\pi^{ij}(t) = \frac{\pi_\theta(a_t|s_t, z_j)}{\pi_\theta(a_t|s_t, z_i)}$ is the importance weight of adapted policy, $\rho_d^{ij}(t) = \frac{p_j(s_{t+1}|s_t, a_t)}{p_i(s_{t+1}|s_t, a_t)}$ denotes the importance weight of transition distributions incurred by dynamics discrepancies, and $Q_\theta(s, a_t, z_j)$ denotes the $Q$-value function for task $\mathcal{T}_j$ fitted by the critic network with parameter $\theta$, which is then used to compute the fitted state value function $V_\theta(s, z_j)$. In meta-RL, the adapted policy for each task can be explicitly known, thus allowing access to the true policy importance weight $\rho_\pi^{ij}(t)$. Referring to Appendix A.2, it is then easy to verify that $V_{ij}^{DR}(s_t = s)$ still holds the doubly robust property. Namely, $V_{ij}^{DR}(s_t = s)$ is unbiased when either the importance weight or value function is correctly estimated. Plugging the unbiasedness of $V_{ij}^{DR}(s_t = s)$ to Eq. (4), the MSE of this DR estimator can thus be simplified to $\mathbb{E}_{\tau_i|_{t:T}} \left[ \left( V_j(s_t) - V_{ij}^{DR}(s_t) \right)^2 | s_t \right] = Var \left( V_{ij}^{DR}(s_t) \right)$, which is dominated by its variance that can be further determined with the following theorem.

**Theorem 3.1** *The variance of estimator $V_{ij}^{DR}(s_t = s)$ can be recursively given by: $\forall t = 1, \cdots, T$*

$$Var_t \left[ V_{ij}^{DR}(s_t = s) \right] = \mathbb{E}_t \left[ (\rho_\pi^{ij}(t))^2 Var_t \left[ r_j(s_t, a_t)|a_t \right] \Big| s_t \right] + Var_t \left[ \rho_\pi^{ij}(t)\Delta(s_t, a_t) \Big| s_t \right]$$

$$+ \mathbb{E}_t \left[ \left( \gamma\rho_\pi^{ij}(t) \left( \rho_d^{ij}(t)V_{ij}^{DR}(s_{t+1}) - \mathbb{E}_{t+1}[V_j(s_{t+1})] \right) - \rho_\pi^{ij}(t)\Delta(s_t, a_t) + V_\theta(s_t, z_j) \right)^2 \right]$$

$$- \mathbb{E}_t \left[ \left( -\rho_\pi^{ij}(t)\Delta(s_t, a_t) + V_\theta(s_t, z_j) \right)^2 \right], \tag{6}$$

*where $Var_t \left[ V_{ij}^{DR}(s_T) \right] = 0$, $\Delta(s_t, a_t) = Q_\theta(s_t, a_t, z_j) - Q_\pi^j(s_t, a_t)$, and $Q_\pi^j$ is the true $Q$-value.*

On the RHS of Eq. (6), the first, second and fourth terms all take expectation over $\pi_\theta(a_t|s_t, z_i)$, while the second variance term is also due to randomness of $\pi_\theta(a_t|s_t, z_i)$. The variance inside the first term stems from randomness of the rewards (which will become zero if the reward function is deterministic), while the expectation inside the third term considers randomness of the future. Theorem 3.1 implies that when the other variables are kept unchanged, a possibly larger value for the

estimate of dynamics importance weight $\rho_d^{ij}$ will enlarge the third term in Eq. (6), which in turn results in a higher variance and consequently a larger MSE. As an empirical validation, in Fig. 1, we show the standard deviation (SD) of state value estimate in the Point-Robot environment by directly using the DR estimator in Eq. (5). It can be seen that the estimated value by DR oscillates significantly during the entire training process, which will eventually cause an unstable and unsuccessful meta-training.

## 3.3 DR Augmented Estimator (DRaE) with Optimal $\hat{\rho}_d^{ij*}$ to Balance Bias and Variance

To overcome the high variance and large MSE issue suffered by the original unbiased DR estimator, we consider the MSE minimization for sample transfer as in Eq. (3), which is equivalent to jointly optimizing the bias and variance w.r.t. the dynamics importance weight $\rho_d^{ij}$. Since the actual transition probability distributions of different tasks are usually inaccessible, in practice, we will use the estimated value of dynamics importance weight $\hat{\rho}_d^{ij}$ instead of its true value $\rho_d^{ij}$. However, by approximating $\rho_d^{ij}$ with $\hat{\rho}_d^{ij}$, the original

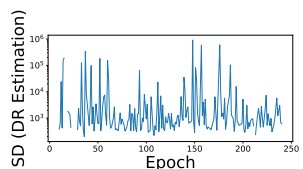

Figure 1: SD of value estimated by direct DR estimator.

unbiased DR estimator $V_{ij}^{DR}(s_t = s)$ in Eq. (5) becomes biased, which is denoted as the DR augmented estimator $\tilde{V}_{ij}^{DR}(s_t = s)$, with the bias given by:

$$\text{Bias}\left(\hat{\rho}_d^{ij}(t)\right) = \left|\mathbb{E}_{a_t \sim \pi_i}\mathbb{E}_{s_{t+1} \sim p_i}\left[\gamma\rho_\pi^{ij}(t)\left(\hat{\rho}_d^{ij}(t)\tilde{V}_{ij}^{DR}(s_{t+1}) - \rho_d^{ij}(t)V_{ij}^{DR}(s_{t+1})\right)\right]\right|. \tag{7}$$

Referring to Appendix A.5 for the detailed derivation, through bounding this $\text{Bias}(\hat{\rho}_d^{ij})$ and by further decomposing the MSE into the sum of bias and variance-related terms, we can obtain an upper bound for the MSE of the biased DR augmented estimator $\tilde{V}_{ij}^{DR}(s_t = s)$:

$$\text{MSE}(\tilde{V}_{ij}^{DR}(s_t = s)) \le \mathbb{E}_t\left[\gamma\rho_\pi^{ij}(t)\left(\hat{\rho}_d^{ij}(t)\tilde{V}_{ij}^{DR}(s_{t+1}) - \rho_d^{ij}(t)V_{ij}^{DR}(s_{t+1})\right)\right]^2 + \left(\mathbb{E}_t V_j(s_t)\right)^2 + \mathbb{V}(\rho_\pi)$$

$$+ \mathbb{E}_t\left[\left(\rho_\pi^{ij}(t)\hat{\rho}_d^{ij}(t)\gamma\tilde{V}_{ij}^{DR}(s_{t+1}) - \rho_\pi^{ij}(t)\Delta(s_t, a_t) + \bar{V}_\theta(s_t, z_j) - \rho_\pi^{ij}(t)\gamma\mathbb{E}_{t+1}[V_j(s_{t+1})]\right)^2\right], \tag{8}$$

where $\mathbb{V}(\rho_\pi)$ denotes the terms on the RHS of Eq. (6) that contain $\rho_\pi^{ij}$ but without $\hat{\rho}_d^{ij}$. We then minimize this upper bound w.r.t. $\hat{\rho}_d^{ij}$ for the MSE reduction, which leads to a convex optimization problem. By letting the first-order derivative of the upper bound in Eq. (8) w.r.t. $\hat{\rho}_d^{ij}$ equal to zero, we can thus obtain the optimal estimated value of dynamics importance weight, as:

$$\hat{\rho}_d^{ij*}(t) = \left(\gamma V_j(s_{t+1}) - r_j(s_t, a_t)\right) \Big/ \left(2\gamma\tilde{V}_{ij}^{DR}(s_{t+1})\right). \tag{9}$$

## 3.4 Interval-Based Approximation for Optimal $\hat{\rho}_d^{ij*}$ of DRaE

Though the optimal $\hat{\rho}_d^{ij*}$ of DRaE can effectively balance the bias and variance by minimizing the MSE, its computation is nontrivial in practice, since the true state value $V_j(s_{t+1})$ at a certain state is also infeasible to access. Here, we propose an interval-based approximation for the optimal $\hat{\rho}_d^{ij*}$, and show that it falls with an interval between $\hat{\rho}_d^{var}$ that achieves the minimum variance in Eq. (6) and the true importance weight $\rho_d^{ij}$. We further verify that any other values of $\hat{\rho}_d^{ij}$ outside this interval $[\hat{\rho}_d^{var}, \rho_d^{ij}]$ will simultaneously enlarge the bias and variance, and consequently result in a larger MSE.

**1) Lower bound $\hat{\rho}_d^{var}$.** The variance of DRaE $\tilde{V}_{ij}^{DR}(s_t)$ can be determined by Eq. (6), with the true $\rho_d^{ij}(t)$ estimated with $\hat{\rho}_d^{ij}(t)$. Through minimizing this variance w.r.t. $\hat{\rho}_d^{ij}(t)$, we formulate a convex optimization problem since the objective is a quadratic function of $\hat{\rho}_d^{ij}(t)$, with the optimal solution $\hat{\rho}_d^{var}(t)$ obtained by letting the first-order derivative of this objective equal to zero:

$$\hat{\rho}_d^{var}(t) = \left(\rho_\pi^{ij}(t)\gamma\mathbb{E}_{t+1}[V_j(s_{t+1})] - V_j(s_t)\right) \Big/ \left(\gamma\rho_\pi^{ij}(t)\tilde{V}_{ij}^{DR}(s_{t+1})\right). \tag{10}$$

The distance from the optimal $\hat{\rho}_d^{ij*}(t)$ to this lower bound $\hat{\rho}_d^{var}(t)$ can then be determined as:

$$\hat{\rho}_d^{ij*}(t) - \hat{\rho}_d^{var}(t) = \left(2V_j(s_t) - \rho_\pi^{ij}(t)\gamma\mathbb{E}_{t+1}[V_j(s_{t+1})] - \rho_\pi^{ij}(t)r(s_t, a_t)\right) \Big/ \left(2\gamma\rho_\pi^{ij}(t)\tilde{V}_{ij}^{DR}(s_{t+1})\right).$$

which will be reduced by increasing the value of policy importance weight $\rho_\pi^{ij}(t)$. In meta-RL, informative trajectories for the target task $\mathcal{T}_j$ are actually sampled from an original task $\mathcal{T}_i$ using $\pi_i$, so the actions in these trajectories have a relatively larger sampling probability by $\pi_i$ than $\pi_j$, resulting in a small $\rho_\pi^{ij}(t)$ especially at the early stage of training. Then, as the training proceeds and with the increase of reward that policy $\pi_j$ can achieve for task $\mathcal{T}_j$, these selected trajectories from task $\mathcal{T}_i$ will be less informative as it is more likely for policy $\pi_j$ to generate them also in the target task $\mathcal{T}_j$. This may lead to an increase of $\rho_\pi^{ij}(t)$ along the training process, which in turn can consistently reduce the distance between $\hat{\rho}_d^{ij*}(t)$ and $\hat{\rho}_d^{var}(t)$. In Section 5.3, we will empirically verify that $\rho_\pi^{ij}(t)$ will generally increase from the initial small values that are much less than one to a stable value around one, which can lead to a tighter lower bound $\hat{\rho}_d^{var}(t)$. In addition, decreasing the estimated value of $\hat{\rho}_d^{ij}(t)$ from the optimal $\hat{\rho}_d^{ij*}(t)$ to the lower bound $\hat{\rho}_d^{var}(t)$ will monotonically reduce the variance but increase the bias of DRaE, while a smaller estimate $\hat{\rho}_d^{ij}(t) < \hat{\rho}_d^{var}(t)$ will cause a consistently larger MSE. Therefore, the optimal $\hat{\rho}_d^{ij*}(t)$ is lower bounded by $\hat{\rho}_d^{var}(t)$.

**2) Upper bound $\rho_d^{ij}$.** Note that when $\hat{\rho}_d^{ij}(t) = \rho_d^{ij}(t)$, our DRaE $\tilde{V}_{ij}^{DR}(s_t)$ becomes the unbiased DR estimator $V_{ij}^{DR}(s_t)$, while any deviation from the true dynamics importance weight $\rho_d^{ij}(t)$ will result in a biased estimator. On the other hand, increasing the value of $\hat{\rho}_d^{ij}(t)$ from $\hat{\rho}_d^{var}(t)$ to $\rho_d^{ij}(t)$ will consistently enlarge the variance. Therefore, a larger estimated value $\hat{\rho}_d^{ij}(t) > \rho_d^{ij}(t)$ will enlarge both the bias and variance, which then indicates that the optimal $\hat{\rho}_d^{ij*}(t)$ is upper bounded by $\rho_d^{ij}(t)$.

In conclusion, we show that the optimal dynamics importance weight $\hat{\rho}_d^{ij*}(t)$ falls within the interval $[\hat{\rho}_d^{var}(t), \rho_d^{ij}(t)]$, while an estimated value of $\hat{\rho}_d^{ij}(t)$ inside this interval can generally achieve a bias-variance tradeoff. In the following, we further derive a sample-independent upper bound to constrain the MSE difference between taking $\hat{\rho}_d^{ij*}(t)$ and $\hat{\rho}_d^{ij}(t)$ as the estimated value in DRaE, respectively.

**Proposition 3.2** *For an estimated dynamics importance weight $\hat{\rho}_d^{ij}(t) \in \left[\hat{\rho}_d^{var}(t), \rho_d^{ij}(t)\right]$, the MSE difference between DRaE taking $\hat{\rho}_d^{ij*}(t)$ and $\hat{\rho}_d^{ij}(t)$ as estimated importance weight is bounded by:*

$$
\left| MSE(\tilde{V}_{ij}^{DR}, \hat{\rho}_d^{ij*}(t)) - MSE(\tilde{V}_{ij}^{DR}, \hat{\rho}_d^{ij}(t)) \right| \leq \frac{3}{4} \left| \mathbb{E}_{a_t \sim \pi_j} \mathbb{E}_{s_{t+1} \sim p_j} \left[ \left( \frac{1}{\rho_d^{ij}(t)} - 2 \right) \gamma V_j(s_{t+1}) \right] \right.
$$

$$
\left. - \mathbb{E}_{a_t \sim \pi_j} \left[ r_j(s_t, a_t) \right] \right|^2 + \max \left( \mathbb{E}_t \left[ \left( A\hat{\rho}_d^{ij*}(t) - B \right)^2 \right], \left| Var_t(\tilde{V}^{DR}, \hat{\rho}_d^{ij*}(t)) - Var_t(V^{DR}, \rho_d^{ij}(t)) \right| \right),
$$

*where $A = \gamma \rho_\pi^{ij}(t) \tilde{V}_{ij}^{DR}(s_{t+1})$ and $B = -\gamma \rho_\pi^{ij}(t) \mathbb{E}_{t+1}[V_j(s_{t+1})] - \rho_\pi^{ij}(t)\Delta(s_t, a_t) + V_\theta(s_t, z_j)$.*

It thus indicates that by setting an upper bound for $\hat{\rho}_d^{ij}(t)$ smaller than the true $\rho_d^{ij}(t)$, we can further reduce the MSE difference between using $\hat{\rho}_d^{ij}(t)$ and $\hat{\rho}_d^{ij*}(t)$. See Appendix A.8 for detailed proof.

## 4 Doubly Robust Augmented Transfer (DRaT) for Meta-RL

Based on our theoretical findings of the MSE reduction for sample transfer by DRaE, we formally present the doubly robust augmented transfer (DRaT) algorithm for meta-RL, and summarize its detailed implementation on PEARL [4] in Algorithm 1. At the data collection phase, for a task $\mathcal{T}_i$, we additionally store the distribution parameters of adapted policy $\pi_i$ that collects data into buffer $\mathcal{B}_i$. Later at the training phase, we sample an informative trajectory set $\mathcal{D}_j$ from all the replay buffers using strategy $\mathcal{S}_I$, which selects trajectories with higher relabeled sum of rewards for task $\mathcal{T}_j$. Hence, any of the hindsight-based methods can be used to determine $\mathcal{S}_I$ here. While for environment where tasks share the same reward, we can simply choose the higher reward trajectories without relabeling.

We then compute the importance weights $\rho_\pi^{ij}$ and $\hat{\rho}_d^{ij}$ for each trajectory in $\mathcal{D}_j$ between the target task $\mathcal{T}_j$ and originally sampled task $\mathcal{T}_i$ as in Line 10. We can obtain $\pi_j$ for task $\mathcal{T}_j$ following the same adaptation procedure as in PEARL, hence $\rho_\pi^{ij}$ of actions in trajectories can be computed as the ratio of sampled probabilities $\pi_j$ and $\pi_i$. While for the dynamics importance weight, a straightforward way to estimate $\hat{\rho}_d^{ij}$ is to predict the distribution of transition $(s, a, s') \in \tau_i \sim \mathcal{D}_j$ by $p_\psi$ and compute it by $\hat{\rho}_d^{ij} = p_\psi^j(\cdot|s, a)/p_\psi^i(\cdot|s, a)$. However, the incorrect estimate of transition distribution in the original task $\mathcal{T}_i$ may cause a small value of $p_\psi^i(\cdot|s, a)$ and thus a large value of $\hat{\rho}_d^{ij}$, possibly frustrating

---

**Algorithm 1** Doubly Robust augmented Transfer (DRaT) for Meta-RL

---

1: **Require:** Training tasks set $\mathcal{T} = \{\mathcal{T}_i\}_{i=1\cdots N}$ from $P_{\mathcal{T}}$, informative data selection strategy $\mathcal{S}_I$
2: Initialize replay buffers $\mathcal{B}_i$ and context $c_i$ for each training task $\mathcal{T}_i$, actor $\pi_\theta$, critic $Q_\theta$, and dynamics prediction network $p_\psi^i(\cdot|s, a)$.
3: **while** meta-training iteration **do**
4:     **for** each training task $\mathcal{T}_i$ **do**
5:         Collect trajectories using policy $\pi_i = \pi_\theta(a|s, z_i)$, $z_i \sim q_\phi(z|c_i)$
6:         Add trajectories and action selection probability of $\pi_i$ to corresponding buffer $\mathcal{B}_i$
7:     **for** each task $\mathcal{T}_j$ **do**
8:         Sample informative trajectory set $\mathcal{D}_j$ and corresponding policy $\pi_i$ using strategy $\mathcal{S}_I$
9:         Sample context $c_j \sim \mathcal{B}_j$ and its latent variable $z_j \sim q_\phi(\cdot|c_j)$, then $\pi_j = \pi_\theta(\cdot|s, z_j)$
10:        Compute importance weights $\rho_\pi^{ij} = \frac{\pi_\theta(a|s, z_j)}{\pi_\theta(a|s, z_i)}$ and $\hat{\rho}_d^{ij} = \max(\frac{p_\psi^j(\cdot|s, a)}{\mathcal{N}(\cdot, \sigma)}, \hat{\rho}_d^l)$
11:        Compute $\tilde{V}^{DR}$ using Eq. (5) with $\rho_\pi^{ij}$ and $\hat{\rho}_d^{ij}(t)$
12:        Compute critic loss in Eq. (1) using $\tilde{V}^{DR}$ as target value and update prediction network
13:     Update the lower bound $\hat{\rho}_d^l$ for clipping $\hat{\rho}_d(t)$

---

the entire training process. Instead, we estimate $\hat{\rho}_d^{ij}$ as guided by the interval-based approximation for optimal $\hat{\rho}_d^{ij*}$ of DRaE in Section 3.4, which is constrained within an interval $[\hat{\rho}_d^{var}, \rho_d^{ij}]$. We predict the distribution of transition $(s, a, s')$ in task $\mathcal{T}_j$ by prediction network $p_\psi^j(\cdot|s, a)$ and in the original task $\mathcal{T}_i$ by Gaussian distribution $\mathcal{N}(s', \sigma)$, which indicates a high probability of transition $(s, a, s') \in \tau^i$ occurring in task $\mathcal{T}_i$. Hence, the maximum value of $\hat{\rho}_d^{ij}$ computed by $p_\psi^j(\cdot|s, a)/\mathcal{N}(\cdot, \sigma)$ will automatically fall into interval $[0, \rho_d^{ij})$ with a high probability, due to the large denominator resulted from the high belief we hold about the transition's occurrence in task $\mathcal{T}_i$. In addition, we also clip the small value of $\hat{\rho}_d$ using $\hat{\rho}_d^l$, which is updated as the average value of $\hat{\rho}_d^{var}$ at this epoch.

The importance weights $\rho_\pi^{ij}$ and $\hat{\rho}_d^{ij}$ are used to estimate the state value in $\mathcal{D}_j$ by DRaE $\tilde{V}^{DR}$ as in Eq. (5), which is used as the target value to compute critic loss in Eq. (1). Finally, the prediction network $p_\psi$ is updated by optimizing the prediction error, and other networks are updated following the same procedure as in PEARL. It is noted that DRaT can also be implemented on other meta-RL baselines, such as MAML [12] and its variants, where the DRaE $\tilde{V}^{DR}$ can be used to compute policy gradients for the meta-policy parameter optimization.

## 5 Experiments

In this section, we assess performance of our DRaT approach following the same evaluation procedure as in [4]. For each benchmark environment, we construct a test task set that is disjoint with the training task set. At the end of each training epoch, we run the evaluation procedure, where trajectories of a fixed length are sampled by the adapted meta-policy from the updated context with continuously sampled transitions. Specifically, we conduct experiments to answer the following questions.

- Can DRaT achieve a better adaptation performance on meta-test tasks in the extreme challenging sparse-reward environment with varying reward functions and dynamics, as compared to hindsight relabeling methods that inherently considers only the difference of reward functions?

- Can DRaT effectively improve the adaptation performance on the standard meta-RL benchmarks where tasks share the identical dense reward function but only differ in distinct dynamics?

- Will the policy importance weight $\rho_\pi^{ij}$ increase along the training process in consistency to the performance improvement of DRaT, which thus provides a tighter lower bound for $\hat{\rho}_d^{ij}$? Will DRaE provide a better value estimate for meta-RL?

**Environment setup.** We compare our DRaT with other baseline algorithms on six robotic control tasks, which can be divided into two families according to their reward and dynamics setup. For the implementation detail of DRaT, please refer to Appendix B.1. These tasks are all simulated via MuJoCo [13], where we further generate various dynamics by randomly sampling the environment parameters, including body mass, body inertia, joint damping, and body component's friction. One

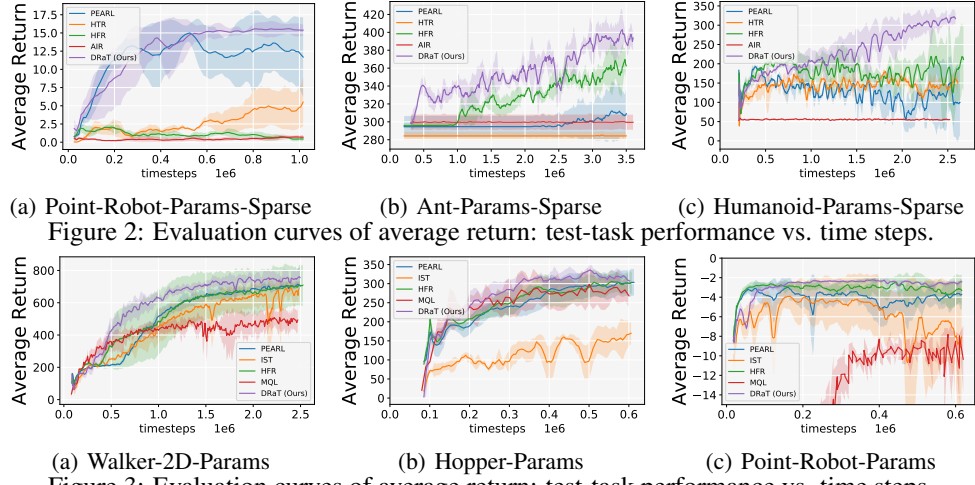

(a) Point-Robot-Params-Sparse    (b) Ant-Params-Sparse    (c) Humanoid-Params-Sparse

Figure 2: Evaluation curves of average return: test-task performance vs. time steps.

(a) Walker-2D-Params    (b) Hopper-Params    (c) Point-Robot-Params

Figure 3: Evaluation curves of average return: test-task performance vs. time steps.

family contains the sparse-reward environments with varying reward functions and dynamics: 1) *Point-Robot-Goal-Params*: control the arm of a 3D robot to reach random goals in the 3D space; 2) *Ant-Goal-Params-Sparse*: navigate a quadruped robot to reach randomly generated goals along the perimeter of a semi-circle; 3) *Humanoid-Goal-Params-Sparse*: navigate a humanoid robot to reach randomly generated goals on a semi-circle. These environments are with the sparse-reward setting in the sense that the RL agent can receive a meaningful reward signal that indicates its distance from the goal only when it reaches areas near the goals. The other family includes environments that have varying dynamics and the identical dense reward function: 1) *Hopper-Params*: control a planar monopod robot to hop as fast as possible and avoid falling; 2) *Walker-2D-Params*: control a 2D bipedal robot to walk and perform the same task as in Hopper; 3) *Point-Robot-Params*: the same as Point-Robot-Goal-Params but with dense reward function. Please refer to Appendices B.2 and B.3 for the detail of varying dynamics and reward function settings, respectively.

### 5.1 Evaluation on Sparse-Reward Environments with Varying Rewards and Dynamics

To answer the first question, we conduct experiments on the first family of sparse-reward environments with varying reward functions and dynamics. We compare DRaT with four baselines: PEARL [4], Hindsight Task Relabeling (HTR) [14], Hindsight Foresight Relabeling (HFR) [15], and Approximate Inverse RL Relabeling (AIR) [6]. Specifically, PEARL is utilized as the baseline meta-RL algorithm for all the other algorithms. For a fair comparison, we keep all the hyper-parameters about meta-RL the same as those in PEARL's source code. In HTR, we implement the single episode relabeling strategy, where hindsight task goal is selected as a state that is reached in a sampled episode and networks in HTR are trained by transitions from that episode which is relabeled by the hindsight goal. HFR utilizes a utility function (foresight) to judge the informativeness of trajectories to a task, and trajectories with higher utilities in certain tasks are more likely to be transferred through the relabelling with reward function (hindsight) to this task. In AIR, a trajectory is relabeled to a new task, for which this trajectory beats the most previously sampled trajectories in terms of the total reward among a sampled candidate task set. We use the same relabeling strategy in both the AIR and our DRaT for selecting informative trajectories for each training task.

**Evaluation results.** We report the averaged return of trajectories sampled in the evaluation phase. From the evaluation return curves of the three tasks as shown in Figs. 2(a)-2(c), our DRaT significantly outperforms the other baseline algorithms, especially in the Ant and Humanoid environments that have higher dimensions of states. In comparison, PEARL struggles to train the Ant and Humanoid under such a challenging environment setting. PEARL-based HTR cannot even learn to move to goals for Ant in such a short period of time steps, while it achieves a similar performance to PEARL for Humanoid. PAERL-based HFR tends to outperform PEARL and other relabeling methods for both Ant and Humanoid, which is possibly because HFR considers relabeling w.r.t. performance of the adapted policy using trajectory instead of just considering the sum of relabeled rewards. We also observe that simply applying AIR into PEARL will even degrade the performance of PEARL. Providing a more accurate estimate of state value functions for transferred informative trajectories, our DRaT generally achieves a better asymptotic performance and can learn faster compared to the other baselines. Note also that in Fig. 2, the solid curve illustrates the mean return on all runs of each algorithm with different random seeds, while the shaded area shows the standard deviation.

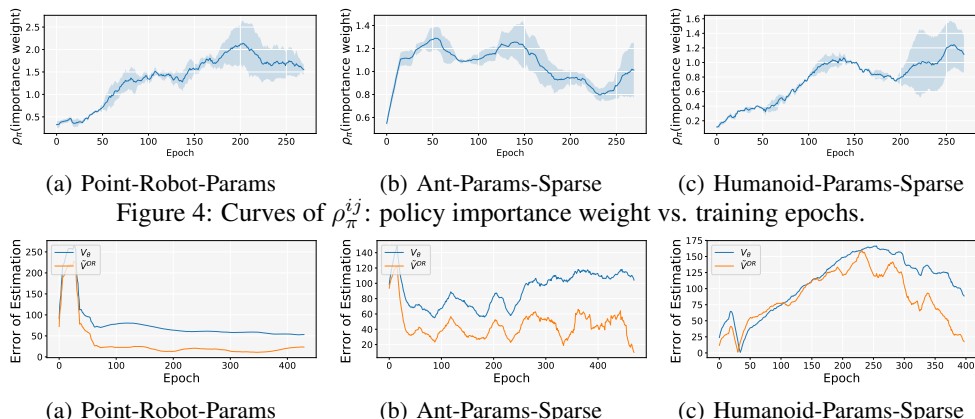

(a) Point-Robot-Params        (b) Ant-Params-Sparse        (c) Humanoid-Params-Sparse

Figure 4: Curves of $\rho_\pi^{ij}$: policy importance weight vs. training epochs.

(a) Point-Robot-Params        (b) Ant-Params-Sparse        (c) Humanoid-Params-Sparse

Figure 5: Curves of errors of value estimation: absolute difference vs. training epochs.

## 5.2 Evaluation on Dense-Reward Environments with Varying Dynamics Only

Then, to answer the second question, we conduct experiments on the second family of environments with varying dynamics and the identical dense reward function. We compare DRaT with four baselines: PEARL [4], Importance Sampling augmented Transfer (IST), Meta-Q-Learning (MQL) [16], and Hindsight Foresight Relabeling (HFR) [14]. Specifically, PEARL is utilized as the baseline meta-RL algorithm for DRaT, IST and HFR. In MQL, for a given transition and task, a fitted logistic classifier outputs probability of the transition being sampled from the task, from which the importance weight can be estimated. Transitions from other tasks are used to compute the meta-RL objective multiplied by this importance weight on a new task, hence the contribution of each transition to the update of meta-parameters is judged by similarity between the new task and its source task. In IST, we use the IS estimator instead of our DRaE, while keeping all the other settings the same as in DRaT.

**Evaluation results.** We report the averaged return of trajectories sampled in the evaluation phase at each epoch, and show the evaluation return curves of the three tasks, respectively, in Figs. 3(a)-3(c). For MQL, we focus specifically on its performance with transfer by the estimated importance weights. Hence, we report its evaluation returns of adapted meta-policy with the aid of transferred samples. It can be seen that with informative samples transferred by DRaT, we can improve the evaluation average return over the other baselines in all the benchmark environments. In addition, DRaT consistently accelerates the training process of Walker-2D-Params and Hopper-Params at the early stage, and also reaches the highest return in Point-Robot-Params. In addition, DRaT presents in general a smaller standard deviation.

## 5.3 Performance Evaluation of Proposed DRaE

We then experimentally estimate the value of $\rho_\pi^{ij}$ to evaluate its trend during the training process of DRaT. We compute the averaged value of $\rho_\pi^{ij}$ estimated for informative trajectories at each training epoch, and show curves of the mean value of Point-Robot-Params, Ant-Params-Sparse, and Humanoid-Params-Sparse in Figs. 4(a)-4(c), respectively, where the shaded area shows the standard deviation on several runs. The results show that value of $\rho_\pi^{ij}$ increases with the improvement of performance on all the training tasks, since selected informative trajectories are more likely to be sampled on target task $\mathcal{T}_j$, which can generate a tighter lower bound as we state in Section 3.4.

At last, we evaluate the estimation error of our proposed DRaT compared to the value network $V_\theta$ as used in PEARL. For a better illustration, we do not plot results of directly using DR estimator, since it may result in a huge variance and estimation error as we analyze in Section 3.2. We analyze the error by computing the absolute difference between the averaged estimated value and averaged true return at each training epoch, and show the error curves of Point-Robot-Params, Ant-Params-Sparse, and Humanoid-Params-Sparse in Figs. 5(a)-5(c), respectively. The results show that our DRaE provides a better value estimation during the training process. It is also observed that curves of DRaT and $V_\theta$ present a similar trend, which is because the computation of DRaE depends on the value of $V_\theta$.

## 6 Computational Complexity Analysis

Additional computational complexity is brought by DRaT for training of dynamics prediction network for each task, computation of DR estimate, and computation of relabelling for informative trajectories. Here, we analytically show that this additional computational complexity is comparable to its baseline

PEARL with a linear scaling factor. Typically in an off-policy meta-RL algorithm, each epoch (i.e., meta-training iteration) is divided into sampling and training phases. In the sampling phase, computational cost stems mainly from the actions chosen by feed-forward computation of the policy network and inference networks. Since all the algorithms (i.e., HTR, HFR, AIR, DRaT) follow the same sampling process as PEARL, we only analyze the computational cost in the training phase.

**PEARL:** In the training phase, the computational cost contains the feed-forward and back-propagation computation of policy network, value network and inference network. Given $K$ training iterations at each epoch, batch size $N_{\mathcal{B}}$ of transitions from $N$ training tasks, state space cardinality $|\mathcal{S}|$ and action space cardinality $|\mathcal{A}|$, and assuming a constant computational cost of feed-forward and back-propagation computation $c_1$, the total computational cost of training phase is $O(K \cdot N_{\mathcal{B}} \cdot N \cdot |\mathcal{S}| \cdot |\mathcal{A}| \cdot c_1)$.

**DRaT:** Besides the same training cost $O(K \cdot N_{\mathcal{B}} \cdot N \cdot |\mathcal{S}| \cdot |\mathcal{A}| \cdot c_1)$ as PEARL, additional computational cost brought by DRaT includes training of dynamics prediction networks, computation of DR estimator, and computation of relabelling, as follows. 1) Considering building separate prediction networks for $N$ training tasks and the cost of feed-forward and back-propagation computation $c_1$, using the same batch of transitions for training, the additional cost of training dynamics prediction networks is also $O(K \cdot N_{\mathcal{B}} \cdot N \cdot |\mathcal{S}| \cdot |\mathcal{A}| \cdot c_1)$. 2) Considering sampling informative trajectories with a maximum length of $L$ for $N$ training tasks and the cost $c_2$ of DR estimation computation at each time step, the computational cost of DR estimation is $O(K \cdot N \cdot L \cdot |\mathcal{S}| \cdot |\mathcal{A}| \cdot c_2)$. 3) We use the approximate inverse RL relabeling (AIR), where we sample one trajectory from each training task for relabeling, leading to $N$ candidate trajectories in total. Assuming that the cost of computing relabeled reward at each time step is $c_3$, the cost of relabeling is then $O(K \cdot N \cdot L \cdot |\mathcal{S}| \cdot |\mathcal{A}| \cdot c_3)$. Thus, the additional computational complexity of DRaT is dominated by $O\Big(K \cdot N \cdot \max\{N_{\mathcal{B}}, L\} \cdot |\mathcal{S}| \cdot |\mathcal{A}| \cdot \max\{c_1, c_2, c_3\}\Big)$, comparable to PEARL with a linear scaling factor $\frac{\max\{N_{\mathcal{B}}, L\}}{N_{\mathcal{B}}} \cdot \frac{\max\{c_1, c_2, c_3\}}{c_1}$.

**Insights for complexity reduction.** 1) To reduce the cost of training dynamics prediction network, we may consider training a meta-dynamics prediction network, which makes prediction for all the tasks with a single network, and thus eliminates the need of training a separate network for each task. 2) To reduce the cost of DR estimation, we may consider a similar solution in $TD(n)$, which makes a trade-off between $TD(0)$ and Monte-Carlo estimation by using the $n$-step rollout and fitted network.

# 7    Conclusion and Limitations

In this paper, we have proposed a doubly robust augmented estimator (DRaE) to tackle the mismatch of dynamics distributions in sample transfer under a more general sparse-reward meta-RL setting, where dynamics and reward functions can both vary across different tasks. While DRaE was established with an optimal dynamics importance weight by minimizing the MSE between estimated and true values of the value function, we proposed a tractable interval-based approximation that guaranteed to cover the optimum. We further developed a doubly robust augmented transfer (DRaT) algorithm for transferring informative samples across tasks in meta-RL. Experimental evaluation on several MuJoCo locomotion tasks demonstrated the effectiveness of our proposed DRaT algorithm.

The major limitation of our method is that it requires computation of the DR estimate and training of the dynamics prediction network for tasks, which may incur additional computational complexity. Besides, like other hindsight-based methods, the reward function is required for relabeling and selection of informative trajectories, which might be inaccessible in some extreme scenarios.

# 8    Acknowledgement

This work was supported in part by the National Natural Science Foundation of China under Grants 62250055, 61931023, T2122024, 62125109, 61932022, 61972256, 61971285, 61831018, 62120106007, 62320106003, and 62371288, in part by the Program of Shanghai Science and Technology Innovation Project under Grant 20511100100, and in part by the Shanghai Rising-Star Program under Grant 20QA1404600.

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
