# Doubly Robust Augmented Transfer for Meta-Reinforcement Learning

## A. Appendix

### A.1. Related Work

**Meta-Reinforcement Learning (Meta-RL).** With the incorporation of meta-learning, meta-RL enables a fast adaptation in RL problems through the idea of "learning to learn". During meta-training, meta-RL learns an inductive bias from a set of relative training tasks for quickly adapting to some new tasks, given only a small amount of samples at the meta-test time. Current meta-RL methods can be classified in to two categories. One is the gradient-based method, which attempts to use a few number of gradient updates to implement the adaptation on a new task, such as by using policy gradient methods to directly update the policy parameters [12, 17, 18, 19]. The other is the context-based method, which builds up an inference network to infer task-specific latent context variables from the input-sampled experience (i.e., context) of the training tasks. A policy with both state and latent variable as input is also trained to maximize rewards on these training tasks, hence the adaptation is conducted by the latent context inference first, followed by the policy adjustment with the inferred latent context as input. These methods mainly differ in their ways of inference [3, 4, 20]. However, sparse reward remains a challenge in meta-RL, where the sparse reward signals provide only scarce task-relevant information and make meta-training and adaptation extremely difficult.

**Sparse-Reward Meta-RL.** To tackle the sparse reward problem in meta-RL, two main research lines have been developed recently. One directly generates informative samples by exploration [21, 22, 23] or by directly using the demonstration datasets [24]. For instance, training a separate exploration policy by maximizing the information gain or intrinsic rewards to collect samples. The other line follows the technique of relabeling that enables sample reuse across tasks, i.e., learning a task at hand by appropriately reusing the samples generated from other tasks. Compared with sample exploration, sample reuse has several advantages, such as no extra exploration, high sample efficiency, and low sample risk. Following the direction of relabeling, hindsight experience replay (HER) [5] has been studied as one typical method, which is originally designed for the multi-goal setting and relabels a trajectory with a lower reward under its original goal to a goal that has higher reward. Packer *et al.* apply hindsight relabeling for meta-RL, and propose hindsight task relabeling (HTR) to relabel the trajectories sampled from one task to a task which can be accomplished in these trajectories with higher rewards [14]. However, like its application on goal-conditioned tasks, this method can only cope with training tasks with different reward functions that correspond to the goals. Taking a step further than hindsight relabelling, Wan *et al.* introduce additionally foresight relabeling to meta-RL, and propose to relabel trajectories to new tasks with higher post-adaptation rewards [15].

**Doubly Robust Estimator.** Doubly robust (DR) is first presented in statistics [25, 26] and then brought into RL by Jiang *et al.* for policy evaluation [7], which combines the direct learning of dynamics models and importance sampling to provide an unbiased and lower-variance value estimate. The variance of DR in value evaluation can be further reduced by applying lower-variance IS estimator [8, 27] and through learning an more accurate dynamics model [28]. For policy learning in RL, Huang *et al.* derive a general form of policy gradient from DR value estimator [29], whereas a DR off-policy actor-critic method is developed by Xu *et al.* [30]. Kallus *et al.* propose the doubly robust method to find a robust policy that can achieve the near-optimum in the worst case under environment distribution shifts [31]. Similar to our work which aims at optimizing MSE of the DR estimator, Su *et al.* derive a shrinkaged importance weight of policy for bandit problem under the assumption of known importance weights, while we do not have access to the true importance weight of dynamics. Different from these works, we apply doubly robust (DR) to transfer the experience collected across a distribution of tasks, for accelerating the value function learning under a challenging sparse-reward meta-RL setting.

**Transfer Learning for Meta-RL.** Our problem setting partly falls into the area of transfer in RL, which aims to accelerate the learning process in a new target task by transferring knowledge learned from the source tasks. Depending on the knowledge to be transferred, these methods in RL can be roughly divided into classes including sampled transitions [32, 33], learned policies or value networks [34, 35, 36, 37], features [38, 39, 40], and skills [41, 42]. Tirinzoni *et al.* apply importance sampling (IS) to transfer samples from a set of source tasks [32], while multiple IS that has a lower variance is

applied in [33]. Our method implements transition transfer by doubly robust methods, which can be proved to have a lower variance than these IS methods.

**A.2. Decomposition of MSE in Eq. (4) in the main text**

$$
\begin{aligned}
\text{MSE}(\hat{V}) =& \mathbb{E}_{\tau_i|_{t:T}}\left[\left(V_j(s_t) - \hat{V}_j(s_t)\right)^2 \big| s_t = s\right] \\
=& \mathbb{E}_{\tau_i|_{t:T}}\left[(V_j(s_t))^2 - 2V_j(s_t)\hat{V}_j(s_t) + (\hat{V}_j(s_t))^2 \big| s_t\right] \\
=& \mathbb{E}_{\tau_i|_{t:T}}\left[(V_j(s_t))^2 - 2V_j(s_t)\hat{V}_j(s_t) + (\mathbb{E}_{\tau_i|_{t:T}}[\hat{V}_j(s_t)|s_t])^2 \big| s_t\right] + \mathbb{E}_{\tau_i|_{t:T}}\left[(V_j(s_t))^2|s_t\right] - \left(\mathbb{E}_{\tau_i|_{t:T}}[\hat{V}_j(s_t)|s_t]\right)^2 \\
=& \left(V^j(s_t) - \mathbb{E}_{\tau_i|_{t:T}}[\hat{V}^j(s_t)|s_t]\right)^2 + \text{Var}(\hat{V}^j(s_t)) = \text{Bias}(\hat{V})^2 + \text{Var}(\hat{V}^j(s_t))
\end{aligned}
$$

**A.3. Doubly Robust Property for Direct Use of Doubly Robust Estimator**

We show the doubly robust property of the DR estimator for value function in Eq. (5) in the main text, as follows.

**1)** In the first case when the importance weight $\rho_\pi$ and $\rho_d$ are correctly estimated and given the state $s_t$ at time step $t$, taking the expectation on the RHS of Eq. (5) in the main text w.r.t. $a_t$ and $s_{t+1}$, we have

$$
\begin{aligned}
& \mathbb{E}_{\substack{\pi_\theta(a_t|s_t,z_i) \\ p_i(s_{t+1}|s_t,a_t)}}\left[V_\theta(s_t,z_j) + \rho_\pi^{ij}(t)\left[r_j(s_t,a_t) + \rho_d^{ij}(t+1)\gamma V_{ij}^{DR}(s_{t+1}) - Q_\theta(s,a,z_j)\right]\right] \\
=& V_\theta(s_t,z_j) + \mathbb{E}_{\substack{\pi_\theta(a_t|s_t,z_i) \\ p_i(s_{t+1}|s_t,a_t)}}\left[\rho_\pi^{ij}(t)\left(r_j(s_t,a_t) + \rho_d^{ij}(t+1)\gamma V_{ij}^{DR}(s_{t+1}) - Q_\theta(s,a,z_j)\right)\right] \\
=& V_\theta(s_t,z_j) + \mathbb{E}_{\substack{\pi_\theta(a_t|s_t,z_j) \\ p_j(s_{t+1}|s_t,a_t)}}\left[r_j(s_t,a_t) + \gamma V_{ij}^{DR}(s_{t+1}) - Q_\theta(s_t,a_t,z_j)\right] \\
=& \mathbb{E}_{\pi_\theta(a_t|s_t,z_j)}\left[r_j(s_t,a_t) + \gamma\mathbb{E}_{p_j(s_{t+1}|s_t,a_t)}V_{ij}^{DR}(s_{t+1})\right],
\end{aligned}
$$

where the last equality follows $V_\theta(s_t,z_j) = \mathbb{E}_{a_t \sim \pi_\theta(\cdot|s_t,z_j)}[Q_\theta(s_t,a_t,z_j)]$ and reduces to the Bellman equation, which is the correct value for state $s_t$'s value in the target task $j$.

**2)** In the other case when $Q_\theta(s_t,a_t,z_j)$ is a correct estimate of the action-state value, namely,

$$
\hat{Q}(s_t,a_t,z_j) = r_j(s_t,a_t) + \gamma\mathbb{E}_{p_j(s_{t+1}|s_t,a_t)}\left[V_{ij}^{DR}(s_{t+1})\right],
$$

which makes the expectation of the second term in Eq. (5) in the main text become zero, then the remaining non-zero term $V_\theta(s_t,z_j)$ is a proper estimate for the state value since recursively expending $V_{ij}^{DR}$ will result in the definition of $Q$-value function.

**A.4. Variance of biased DR estimator using $\hat{\rho}_d$**

We firstly derive the variance of biased DR estimator $\tilde{V}^{DR}$ using an arbitrary importance weight $\hat{\rho}_d$. Let $\delta = \mathbb{E}_t[\tilde{V}_{ij}^{DR}(s_t) - V^j(s_t)]$ denote the difference between $\tilde{V}_{ij}^{DR}$ and $V^j$, hence the bias of $\tilde{V}_{ij}^{DR}$ by using $\hat{\rho}_d$ can be denoted as $Bias(\hat{\rho}) = |\delta|$. Then, the variance $Var_t[V_{ij}^{DR}(s_t)]$ can be obtained by letting $\hat{\rho}_d = \rho_d$. Given a certain state $s_t$, namely, the distribution is

conditioned on $s_t$, we thus have

$$Var_t\left[\tilde{V}_{ij}^{DR}(s_t)\right]$$

$$=\mathbb{E}_t[\tilde{V}_{ij}^{DR}(s_t)^2] - (\mathbb{E}_t[\tilde{V}_{ij}^{DR}(s_t)])^2$$

$$=\mathbb{E}_t[\tilde{V}_{ij}^{DR}(s_t)^2] - (\mathbb{E}_t[V^j(s_t)] + \mathbb{E}_t[\delta])^2$$

$$=\mathbb{E}_t[\tilde{V}_{ij}^{DR}(s_t)^2] - (\mathbb{E}_t[V^j(s_t)])^2 - 2\mathbb{E}_t V^j(s_t)\mathbb{E}[\delta] - \left(\mathbb{E}[\delta]\right)^2$$

$$=\mathbb{E}_t\left[\left(\bar{V}_\theta(s_t,z_j) + \rho_\pi^{ij}(t)\hat{\rho}_d^{ij}(t)\gamma\tilde{V}_{ij}^{DR}(s_{t+1}) + \rho_\pi^{ij}(t)(r(s_t,a_t) - \bar{Q}_\theta(s_t,a_t,z_j))\right)^2 - V^j(s_t)^2\right]$$

$$+ Var_t\left[V^j(s_t)\right] - 2\mathbb{E}_t V^j(s_t)\mathbb{E}[\delta] - \left(\mathbb{E}[\delta]\right)^2$$

$$=\mathbb{E}_t\left[\left(\bar{V}_\theta(s_t,z_j) + \rho_\pi^{ij}(t)\gamma\tilde{V}_{ij}^{DR}(s_{t+1}) + \rho_\pi^{ij}(t)(r(s_t,a_t) - \bar{Q}_\theta(s_t,a_t,z_j))\right.\right.$$

$$\left.\left. + \rho_\pi^{ij}(t)(\hat{\rho}_d^{ij}(t) - 1)\gamma\tilde{V}_{ij}^{DR}(s_{t+1})\right)^2 - V^j(s_t)^2\right] - 2\mathbb{E}_t V^j(s_t)\mathbb{E}[\delta] - \left(\mathbb{E}[\delta]\right)^2 \tag{A.1}$$

$$=\mathbb{E}_t\left[\left(\rho_\pi^{ij}(t)Q^j(s_t,a_t) - \rho_\pi^{ij}(t)\bar{Q}_\theta(s_t,a_t,z_j) + \bar{V}_\theta(s_t,z_j) + \rho_\pi^{ij}(t)(r(s_t,a_t) + \gamma\tilde{V}_{ij}^{DR}(s_{t+1}) - Q^j(s_t,a_t))\right.\right.$$

$$\left.\left. + \rho_\pi^{ij}(t)(\hat{\rho}_d^{ij}(t) - 1)\gamma\tilde{V}_{ij}^{DR}(s_{t+1})\right)^2 - V^j(s_t)^2\right] - 2\mathbb{E}_t V^j(s_t)\mathbb{E}[\delta] - \left(\mathbb{E}[\delta]\right)^2 \tag{A.2}$$

$$=\mathbb{E}_t\left[\left((-\rho_\pi^{ij}(t)\Delta(s_t,a_t) + \bar{V}_\theta(s_t,z_j)) + \rho_\pi^{ij}(t)(r(s_t,a_t) - R(s_t,a_t)) + \rho_\pi^{ij}(t)\gamma(\tilde{V}_{ij}^{DR}(s_{t+1}) - \mathbb{E}_{t+1}[V^j(s_{t+1})])\right.\right.$$

$$\left.\left. + \rho_\pi^{ij}(t)(\hat{\rho}_d^{ij}(t) - 1)\gamma\tilde{V}_{ij}^{DR}(s_{t+1})\right)^2 - V^j(s_t)^2\right] - 2\mathbb{E}_t V^j(s_t)\mathbb{E}[\delta] - \left(\mathbb{E}[\delta]\right)^2 \tag{A.3}$$

$$=\mathbb{E}_t\left[(-\rho_\pi^{ij}(t)\Delta(s_t,a_t) + \bar{V}_\theta(s_t,z_j))^2 - V^j(s_t)^2\right] + \mathbb{E}_t\left[(\rho_\pi^{ij}(t)(r(s_t,a_t) - R(s_t,a_t))^2\right]$$

$$+ \mathbb{E}_t\left[\left(\rho_\pi^{ij}(t)\gamma(\tilde{V}_{ij}^{DR}(s_{t+1}) - \mathbb{E}_{t+1}[V^j(s_{t+1})])\right)^2\right] + \mathbb{E}_t\left[\left(\rho_\pi^{ij}(t)\left(\hat{\rho}_d^{ij}(t) - 1\right)\gamma\tilde{V}_{ij}^{DR}(s_{t+1})\right)^2\right]$$

$$+ 2\mathbb{E}_t\left[(-\rho_\pi^{ij}(t)\Delta(s_t,a_t) + \bar{V}_\theta(s_t,z_j))(\rho_\pi^{ij}(t)\left(\hat{\rho}_d^{ij}(t) - 1\right)\gamma\tilde{V}_{ij}^{DR}(s_{t+1}))\right]$$

$$+ 2\mathbb{E}_t\left[\rho_\pi^{ij}(t)\gamma(\tilde{V}_{ij}^{DR}(s_{t+1}) - \mathbb{E}_{t+1}[V^j(s_{t+1})])\rho_\pi^{ij}(t)\left(\hat{\rho}_d^{ij}(t) - 1\right)\gamma\tilde{V}_{ij}^{DR}(s_{t+1})\right] - 2\mathbb{E}_t V^j(s_t)\mathbb{E}[\delta] - \left(\mathbb{E}[\delta]\right)^2 \tag{A.4}$$

$$=Var_t\left[-\rho_\pi^{ij}(t)\Delta(s_t,a_t) + \bar{V}_\theta(s_t,z_j)|s_t\right] + \mathbb{E}_t\left[(\rho_\pi^{ij}(t))^2 Var_t\left[r(s_t,a_t)|a_t\right]|s_t\right]$$

$$+ \mathbb{E}_t\left[(\rho_\pi^{ij}(t))^2\gamma^2 Var_{t+1}(\tilde{V}_{ij}^{DR}(s_{t+1})|a_t)|s_t\right] + \mathbb{E}_t\left[\left(\rho_\pi^{ij}(t)\left(\hat{\rho}_d^{ij}(t) - 1\right)\gamma\tilde{V}_{ij}^{DR}(s_{t+1})\right)^2\right]$$

$$+ 2\mathbb{E}_t\left[\left(-\rho_\pi^{ij}(t)\Delta(s_t,a_t) + \bar{V}_\theta(s_t,z_j) + \rho_\pi^{ij}(t)\gamma(\tilde{V}_{ij}^{DR}(s_{t+1}) - \mathbb{E}_{t+1}[V^j(s_{t+1})])\right)\left(\rho_\pi^{ij}(t)(\hat{\rho}_d^{ij}(t) - 1)\gamma\tilde{V}_{ij}^{DR}(s_{t+1})\right)\right]$$

$$+ \mathbb{E}_{a_t}\left[(\rho_\pi^{ij}(t)\gamma)^2 Var_{t+1}\left[(\tilde{V}_{ij}^{DR}(s_{t+1}))|s_t,a_t\right]\right] + \mathbb{E}_{a_t}\left[\left(-\rho_\pi^{ij}(t)\Delta(s_t,a_t) + \bar{V}_\theta(s_t,z_j)\right)^2\right]$$

$$- \mathbb{E}_{a_t}\left[(\rho_\pi^{ij}(t)\gamma)^2 Var_{t+1}\left[(\tilde{V}_{ij}^{DR}(s_{t+1}))|s_t,a_t\right]\right] - \mathbb{E}_{a_t}\left[\left(-\rho_\pi^{ij}(t)\Delta(s_t,a_t) + \bar{V}_\theta(s_t,z_j)\right)^2\right] - 2\mathbb{E}_t V^j(s_t)\mathbb{E}[\delta] - \left(\mathbb{E}[\delta]\right)^2 \tag{A.5}$$

$$=Var_t\left[-\rho_\pi^{ij}(t)\Delta(s_t,a_t) + \bar{V}_\theta(s_t,z_j)|s_t\right] + \mathbb{E}_t\left[(\rho_\pi^{ij}(t))^2 Var_t\left[r(s_t,a_t)|a_t\right]|s_t\right]$$

$$+ \mathbb{E}_t\left[(\rho_\pi^{ij}(t))^2\gamma^2 Var_{t+1}(\tilde{V}_{ij}^{DR}(s_{t+1})|a_t)|s_t\right] + \mathbb{E}_t\left[\left(\rho_\pi^{ij}(t)\left(\hat{\rho}_d^{ij}(t) - 1\right)\gamma\tilde{V}_{ij}^{DR}(s_{t+1})\right.\right.$$

$$\left.\left. - \rho_\pi^{ij}(t)\Delta(s_t,a_t) + \bar{V}_\theta(s_t,z_j) + \rho_\pi^{ij}(t)\gamma(\tilde{V}_{ij}^{DR}(s_{t+1}) - \mathbb{E}_{t+1}[V^j(s_{t+1})])\right)^2\right]$$

$$- \mathbb{E}_{a_t}\left[(\rho_\pi^{ij}(t)\gamma)^2 Var_{t+1}\left[(\tilde{V}_{ij}^{DR}(s_{t+1}))|s_t,a_t\right]\right] - \mathbb{E}_{a_t}\left[\left(-\rho_\pi^{ij}(t)\Delta(s_t,a_t) + \bar{V}_\theta(s_t,z_j)\right)^2\right] - 2\mathbb{E}_t V^j(s_t)\mathbb{E}[\delta] - \left(\mathbb{E}[\delta]\right)^2 \tag{A.6}$$

$$=Var_t\left[\rho_\pi^{ij}(t)\Delta(s_t,a_t)|s_t\right] + \mathbb{E}_t\left[(\rho_\pi^{ij}(t))^2 Var_t\left[r(s_t,a_t)|a_t\right]|s_t\right] + \mathbb{E}_t\left[\left(\rho_\pi^{ij}(t)\hat{\rho}_d^{ij}(t)\gamma\tilde{V}_{ij}^{DR}(s_{t+1})\right.\right.$$

$$\left.\left. - \rho_\pi^{ij}(t)\Delta(s_t,a_t) + \bar{V}_\theta(s_t,z_j) - \rho_\pi^{ij}(t)\gamma\mathbb{E}_{t+1}[V^j(s_{t+1})]\right)^2\right] - \mathbb{E}_{a_t}\left[\left(-\rho_\pi^{ij}(t)\Delta(s_t,a_t) + \bar{V}_\theta(s_t,z_j)\right)^2\right] - 2\mathbb{E}_t V^j(s_t)\mathbb{E}[\delta] - \left(\mathbb{E}[\delta]\right)^2. \tag{A.7}$$

We eliminate $Var_t\left[V^j(s_t)\right]$ in Eq. (A.1) since $Var_t\left[V^j(s_t)\right] = 0$ when $s_t$ is given. The equivalence from Eq. (A.2) to Eq. (A.3) uses the fact that $Q^j(s_t, a_t) = R(s_t, a_t) + \gamma\mathbb{E}_{t+1}[V^j(s_{t+1})]$. The equivalence from Eq. (A.3) to Eq. (A.4) follows the extension of the square of sum of the four terms, namely, the square of the first parentheses in Eq. (A.3) and the following facts given $s_t$ and $a_t$: 1) $r(s_t, a_t) - R(s_t, a_t)$ and $\tilde{V}_{ij}^{DR}(s_{t+1}) - \mathbb{E}_{t+1}[V^j(s_{t+1})]$ are random variables with zero mean and independent of each others, since $R(s_t, a_t)$ and $\mathbb{E}_{t+1}[V^j(s_{t+1})]$ are the mean of $r(s_t, a_t)$ and $\tilde{V}_{ij}^{DR}(s_{t+1})$ respectively ; 2) $r(s_t, a_t) - R(s_t, a_t)$ and $\tilde{V}_{ij}^{DR}(s_{t+1})$ are independent; 3) $(-\rho_\pi^{ij}(t)\Delta(s_t, a_t) + \bar{V}_\theta(s_t, z_j))$ is constant.

The equivalence from Eq. (A.5) to Eq. (A.6) follows:

$$\mathbb{E}_t\left[\left(-\rho_\pi^{ij}(t)\Delta(s_t, a_t) + \bar{V}_\theta(s_t, z_j) + \rho_\pi^{ij}(t)\gamma(\tilde{V}_{ij}^{DR}(s_{t+1}) - \mathbb{E}_{t+1}[V^j(s_{t+1})])\right)^2|s_t\right]$$

$$=\mathbb{E}_{a_t}\left[\mathbb{E}_t\left[\left(-\rho_\pi^{ij}(t)\Delta(s_t, a_t) + \bar{V}_\theta(s_t, z_j) + \rho_\pi^{ij}(t)\gamma(\tilde{V}_{ij}^{DR}(s_{t+1}) - \mathbb{E}_{t+1}[V^j(s_{t+1})])\right)^2|s_t, a_t\right]\right]$$

$$=\mathbb{E}_{a_t}\left[Var_t\left[\left(-\rho_\pi^{ij}(t)\Delta(s_t, a_t) + \bar{V}_\theta(s_t, z_j) + \rho_\pi^{ij}(t)\gamma(\tilde{V}_{ij}^{DR}(s_{t+1}) - \mathbb{E}_{t+1}[V^j(s_{t+1})])\right)|s_t, a_t\right]\right]$$

$$+\mathbb{E}_{a_t}\left[\mathbb{E}_t\left[\left(-\rho_\pi^{ij}(t)\Delta(s_t, a_t) + \bar{V}_\theta(s_t, z_j) + \rho_\pi^{ij}(t)\gamma(\tilde{V}_{ij}^{DR}(s_{t+1}) - \mathbb{E}_{t+1}[V^j(s_{t+1})])\right)|s_t, a_t\right]^2\right] \quad\text{(A.8)}$$

$$=\mathbb{E}_{a_t}\left[(\rho_\pi^{ij}(t)\gamma)^2 Var_t\left[(\tilde{V}_{ij}^{DR}(s_{t+1}))|s_t, a_t\right]\right] + \mathbb{E}_{a_t}\left[\left(-\rho_\pi^{ij}(t)\Delta(s_t, a_t) + \bar{V}_\theta(s_t, z_j)\right)^2\right], \quad\text{(A.9)}$$

where the last step is obtained from the equivalence of the variance and the expectation in Eq. (A.8):

$$Var_t\left[\left(-\rho_\pi^{ij}(t)\Delta(s_t, a_t) + \bar{V}_\theta(s_t, z_j) + \rho_\pi^{ij}(t)\gamma(\tilde{V}_{ij}^{DR}(s_{t+1}) - \mathbb{E}_{t+1}[V^j(s_{t+1})])\right)|s_t, a_t\right]$$

$$=Var_t\left[\rho_\pi^{ij}(t)\gamma(\tilde{V}_{ij}^{DR}(s_{t+1}) - \mathbb{E}_{t+1}[V^j(s_{t+1})])|s_t, a_t\right]$$

$$=(\rho_\pi^{ij}(t)\gamma)^2 Var_t\left[(\tilde{V}_{ij}^{DR}(s_{t+1}))|s_t, a_t\right],$$

$$\mathbb{E}_{a_t}\left[\mathbb{E}_t\left[\left(-\rho_\pi^{ij}(t)\Delta(s_t, a_t) + \bar{V}_\theta(s_t, z_j) + \rho_\pi^{ij}(t)\gamma(\tilde{V}_{ij}^{DR}(s_{t+1}) - \mathbb{E}_{t+1}[V^j(s_{t+1})])\right)|s_t, a_t\right]^2\right]$$

$$=\mathbb{E}_{a_t}\left[\left(-\rho_\pi^{ij}(t)\Delta(s_t, a_t) + \bar{V}_\theta(s_t, z_j)\right)^2\right]. \qquad\text{// use fact 1) above}$$

The equivalence from Eq. (A.6) to Eq. (A.7) is from the fact that $\bar{V}_\theta(s_t, z_j)$ is constant given $s_t$. Given that $Q_\theta = 0$, $\tilde{V}_{ij}^{DR}(s_t)$ will degrade to the variance of IS estimator and its variance can be written as follows:

$$Var_t[\tilde{V}_{ij}^{IS}(s_t)] = Var_t\left[\rho_\pi^{ij}(t)Q_\pi(s_t, a_t)|s_t\right] + \mathbb{E}_t\left[(\rho_\pi^{ij}(t))^2 Var_t\left[r(s_t, a_t)|a_t\right]|s_t\right] + \mathbb{E}_t\left[\left(\rho_\pi^{ij}(t)\hat{\rho}_d^{ij}(t)\gamma\tilde{V}_{ij}^{IS}(s_{t+1})\right.\right.$$

$$\left.\left. -\rho_\pi^{ij}(t)Q_\pi(s_t, a_t) + \bar{V}_\theta(s_t, z_j) - \rho_\pi^{ij}(t)\gamma\mathbb{E}_{t+1}[V^j(s_{t+1})]\right)^2\right] - \mathbb{E}_{a_t}\left[\left(-\rho_\pi^{ij}(t)Q_\pi(s_t, a_t) + \bar{V}_\theta(s_t, z_j)\right)^2\right]. \quad\text{(A.10)}$$

We further denote

$$\mathbb{V}(\rho_\pi) = \mathbb{E}_t\left[(\rho_\pi^{ij}(t))^2 Var_t\left[r_j(s_t, a_t)|a_t\right]\bigg|s_t\right] + Var_t\left[\rho_\pi^{ij}(t)\Delta(s_t, a_t)\bigg|s_t\right] - \mathbb{E}_t\left[\left(-\rho_\pi^{ij}(t)\Delta(s_t, a_t) + V_\theta(s_t, z_j)\right)^2\right],$$

which corresponds to the first, the second, and the third terms in Eq. (A.7).

**A.5. Proof of Theorem 3.1**

In this section, we derive the variance of unbiased DR estimator in Eq. (6) as shown in Theorem 3.1. Letting $\hat{\rho}_d = \rho_d$ in Eq. (A.7), we have $\delta = 0$ and the variance can be obtained as:

$$Var_t[V_{ij}^{DR}(s_t = s)] = Var_t\left[\rho_\pi^{ij}(t)\Delta(s_t, a_t)|s_t\right] + \mathbb{E}_t\left[(\rho_\pi^{ij}(t))^2 Var_t\left[r(s_t, a_t)|a_t\right]|s_t\right] + \mathbb{E}_t\left[\left(\rho_\pi^{ij}(t)\hat{\rho}_d^{ij}(t)\gamma\tilde{V}_{ij}^{DR}(s_{t+1})\right.\right.$$

$$\left.\left. -\rho_\pi^{ij}(t)\Delta(s_t, a_t) + \bar{V}_\theta(s_t, z_j) - \rho_\pi^{ij}(t)\gamma\mathbb{E}_{t+1}[V^j(s_{t+1})]\right)^2\right] - \mathbb{E}_{a_t}\left[\left(-\rho_\pi^{ij}(t)\Delta(s_t, a_t) + \bar{V}_\theta(s_t, z_j)\right)^2\right].$$

**A.6. Upper bound for MSE of biased DR estimator $\tilde{V}^{DR}$**

$$\text{Bias}(\hat{\rho}_d^{ij}) = \left| \mathbb{E}_{a_t \sim \pi_i} \mathbb{E}_{s_{t+1} \sim p_i} \left[ \tilde{V}_{ij}^{DR}(s_t = s) \right] - V^j(s_t = s) \right|$$

$$= \left| \mathbb{E}_{a_t \sim \pi_i} \mathbb{E}_{s_{t+1} \sim p_i} \left[ \gamma \rho_\pi^{ij}(t) \left( \hat{\rho}_d^{ij}(t) \tilde{V}_{ij}^{DR}(s_{t+1}) - \rho_d^{ij}(t) V_{ij}^{DR}(s_{t+1}) \right) \right] \right|, \quad \text{(A.11)}$$

where the second equality is obtained by the unbiasedness of $V_{ij}^{DR}$ to $V^j$. Following the decomposition in Section A.2, MSE of the biased DR estimator $\tilde{V}^{DR}$ can be written as:

$$MSE(\tilde{V}_{ij}^{DR}(s_t = s)) = Var_t \left[ \rho_\pi^{ij}(t) \Delta(s_t, a_t) | s_t \right] + \mathbb{E}_t \left[ (\rho_\pi^{ij}(t))^2 Var_t \left[ r(s_t, a_t) | a_t \right] | s_t \right] + \mathbb{E}_t \left[ \left( \rho_\pi^{ij}(t) \hat{\rho}_d^{ij}(t) \gamma \tilde{V}_{ij}^{DR}(s_{t+1}) \right. \right.$$

$$\left. \left. - \rho_\pi^{ij}(t) \Delta(s_t, a_t) + \bar{V}_\theta(s_t, z_j) - \rho_\pi^{ij}(t) \gamma \mathbb{E}_{t+1}[V^j(s_{t+1})] \right)^2 \right] - \mathbb{E}_{a_t} \left[ \left( - \rho_\pi^{ij}(t) \Delta(s_t, a_t) + \bar{V}_\theta(s_t, z_j) \right)^2 \right] - 2\mathbb{E}_t V^j(s_t) \mathbb{E}[\delta],$$

where the last term can be bounded as

$$-2\mathbb{E}_t V^j(s_t) \mathbb{E}[\delta] \le \left( \mathbb{E}_t V^j(s_t) \right)^2 + \left( \mathbb{E}[\delta] \right)^2 = \left( \mathbb{E}_t V^j(s_t) \right)^2 + \left( Bias(\hat{\rho}_d^{ij}) \right)^2,$$

with the bias bounded according to the Jesen's inequality

$$\text{Bias}(\hat{\rho}_d^{ij}) \le \sqrt{\mathbb{E}_{a_t \sim \pi_i} \mathbb{E}_{s_{t+1} \sim p_i} \left[ \gamma \rho_\pi^{ij}(t) \left( \hat{\rho}_d^{ij}(t) \tilde{V}_{ij}^{DR}(s_{t+1}) - \rho_d^{ij}(t) V_{ij}^{DR}(s_{t+1}) \right) \right]^2}.$$

Hence, we can obtain an upper bound for the MSE of $\tilde{V}_{ij}^{DR}(s_t = s)$:

$$MSE(\tilde{V}_{ij}^{DR}(s_t = s)) \le Var_t \left[ \rho_\pi^{ij}(t) \Delta(s_t, a_t) | s_t \right] + \mathbb{E}_t \left[ (\rho_\pi^{ij}(t))^2 Var_t \left[ r(s_t, a_t) | a_t \right] | s_t \right] + \mathbb{E}_t \left[ \left( \rho_\pi^{ij}(t) \hat{\rho}_d^{ij}(t) \gamma \tilde{V}_{ij}^{DR}(s_{t+1}) \right. \right.$$

$$\left. \left. - \rho_\pi^{ij}(t) \Delta(s_t, a_t) + \bar{V}_\theta(s_t, z_j) - \rho_\pi^{ij}(t) \gamma \mathbb{E}_{t+1}[V^j(s_{t+1})] \right)^2 \right] - \mathbb{E}_{a_t} \left[ \left( - \rho_\pi^{ij}(t) \Delta(s_t, a_t) + \bar{V}_\theta(s_t, z_j) \right)^2 \right]$$

$$+ \mathbb{E}_{a_t \sim \pi_i} \mathbb{E}_{s_{t+1} \sim p_i} \left[ \gamma \rho_\pi^{ij}(t) \left( \hat{\rho}_d^{ij}(t) \tilde{V}_{ij}^{DR}(s_{t+1}) - \rho_d^{ij}(t) V_{ij}^{DR}(s_{t+1}) \right) \right]^2 + \left( \mathbb{E}_t V^j(s_t) \right)^2. \quad \text{(A.12)}$$

Note that $\mathbb{V}(\rho_\pi)$ also denote the terms that contains $\rho_\pi$ but without $\hat{\rho}_d$ in Eq. (A.12).

**A.7. Reduction of MSE by optimizing upper bound w.r.t. $\hat{\rho}_d$**

Optimization of the upper bound in Eq. (A.12) w.r.t. $\hat{\rho}_d$ can be formulated as:

$$\min_{\hat{\rho}_d} \mathbb{E}_t \left[ \gamma \rho_\pi^{ij}(t) \left( \hat{\rho}_d^{ij}(t) \tilde{V}_{ij}^{DR}(s_t) - \rho_d^{ij}(t) V_{ij}^{DR}(s_t) \right) \right]^2 + \mathbb{E}_t \left[ \left( \rho_\pi^{ij}(t) \gamma \left( \hat{\rho}_d^{ij}(t) \tilde{V}_{ij}^{DR}(s_{t+1}) - \mathbb{E}_{t+1}[V^j(s_{t+1})] \right) \right. \right.$$

$$\left. \left. - \rho_\pi^{ij}(t) \Delta(s_t, a_t) + \bar{V}_\theta(s_t, z_j) \right)^2 \right].$$

This optimization problem is convex w.r.t. $\hat{\rho}_d$. By letting the first-order derivative of the objective function be zero, we have:

$$2\mathbb{E}_t \left[ (\gamma \rho_\pi^{ij}(t))^2 \tilde{V}_{ij}^{DR}(s_{t+1}) \left( \hat{\rho}_d^{ij}(t) \tilde{V}_{ij}^{DR}(s_{t+1}) - \rho_d^{ij}(t) V_{ij}^{DR}(s_{t+1}) \right) \right]$$

$$+ 2\mathbb{E}_t \left[ \gamma \rho_\pi^{ij}(t) \tilde{V}_{ij}^{DR}(s_{t+1}) \left( \rho_\pi^{ij}(t) \gamma \left( \hat{\rho}_d^{ij}(t) \tilde{V}_{ij}^{DR}(s_{t+1}) - \mathbb{E}_{t+1}[V^j(s_{t+1})] \right) - \rho_\pi^{ij}(t) \Delta(s_t, a_t) + \bar{V}_\theta(s_t, z_j) \right) \right] = 0.$$

By eliminating the constant of 2 and merging the two expectations on the left-hand side into one expectation, we have:

$$\mathbb{E}_t \left[ \gamma \rho_\pi^{ij}(t) \tilde{V}_{ij}^{DR}(s_{t+1}) \cdot \left( 2\gamma \rho_\pi^{ij}(t) \hat{\rho}_d^{ij}(t) \tilde{V}_{ij}^{DR}(s_{t+1}) \right. \right.$$

$$\left. \left. - \gamma \rho_\pi^{ij}(t) \rho_d^{ij}(t) V_{ij}^{DR}(s_{t+1}) - \gamma \rho_\pi^{ij}(t) \mathbb{E}_{t+1}[V^j(s_{t+1})] - \rho_\pi^{ij}(t) \Delta(s_t, a_t) + \bar{V}_\theta(s_t, z_j) \right) \right] = 0.$$

Note that inside expectation on the left-hand side is the multiplication of $\gamma \rho_\pi^{ij}(t) \tilde{V}_{ij}^{DR}(s_{t+1})$ with a summation enclosed by the parentheses, which can be rewritten as:

$$\left( 2\gamma \rho_\pi^{ij}(t) \hat{\rho}_d^{ij}(t) \tilde{V}_{ij}^{DR}(s_{t+1}) - \gamma \rho_\pi^{ij}(t) \rho_d^{ij}(t) V_{ij}^{DR}(s_{t+1}) - \gamma \rho_\pi^{ij}(t) \mathbb{E}_{t+1}[V^j(s_{t+1})] - \rho_\pi^{ij}(t) \Delta(s_t, a_t) + \bar{V}_\theta(s_t, z_j) \right)$$

$$= \gamma \rho_\pi^{ij}(t) \left( 2\hat{\rho}_d^{ij}(t) \tilde{V}_{ij}^{DR}(s_{t+1}) - \rho_d^{ij}(t) V_{ij}^{DR}(s_{t+1}) - \mathbb{E}_{t+1}[V^j(s_{t+1})] \right) - \rho_\pi^{ij}(t) \Delta(s_t, a_t) + \bar{V}_\theta(s_t, z_j).$$

On the right-hand side of this equation, the first three terms are closely correlated to $\gamma \rho_\pi^{ij}(t) \tilde{V}_{ij}^{DR}(s_{t+1})$, while the last two terms are loosely correlated to it. Furthermore, given that $\hat{\rho}_d^{ij}(t)$ is inside an interval upper-bounded by its true value $\rho_d^{ij}(t)$, the value of $\hat{\rho}_d^{ij}(t) \tilde{V}_{ij}^{DR}(s_{t+1})$ is comparable to those of $\rho_d^{ij}(t) V_{ij}^{DR}(s_{t+1})$ and $\mathbb{E}_{t+1}[V^j(s_{t+1})]$. Thus, the first three terms will be compensated by each other, while value of the last two terms $-\rho_\pi^{ij}(t) \Delta(s_t, a_t) + \bar{V}_\theta(s_t, z_j)$ will dominate, which is loosely correlated to $\gamma \rho_\pi^{ij}(t) \tilde{V}_{ij}^{DR}(s_{t+1})$.

Since $\gamma \rho_\pi^{ij}(t) \tilde{V}_{ij}^{DR}(s_{t+1})$ cannot dominate the value in the parentheses, we make assumption that it is loosely correlated to the term in the parentheses and have

$$2\mathbb{E}_t\left[ \gamma \rho_\pi^{ij}(t) \hat{\rho}_d^{ij}(t) \tilde{V}_{ij}^{DR}(s_{t+1}) \right] - \mathbb{E}_t\left[ \gamma \rho_\pi^{ij}(t) \rho_d^{ij}(t) V_{ij}^{DR}(s_{t+1}) \right] + \mathbb{E}_t\left[ \left( -\rho_\pi^{ij}(t) \gamma \mathbb{E}_{t+1}[V^j(s_{t+1})] - \rho_\pi^{ij}(t) \Delta(s_t, a_t) + \bar{V}_\theta(s_t, z_j) \right) \right] = 0,$$

$$2\mathbb{E}_t\left[ \gamma \rho_\pi^{ij}(t) \hat{\rho}_d^{ij}(t) \tilde{V}_{ij}^{DR}(s_{t+1}) \right] - \mathbb{E}_{a_t \sim \pi_j, s_{t+1} \sim p_j}\left[ \gamma V_{ij}^{DR}(s_{t+1}) \right] - \gamma \mathbb{E}_{a_t \sim \pi_j} \mathbb{E}_{t+1}\left[ V^j(s_{t+1}) \right] + V^j(s_t) = 0,$$

$$2\mathbb{E}_{a_t \sim \pi_j} \mathbb{E}_{s_{t+1} \sim p_i}\left[ \gamma \tilde{V}_{ij}^{DR}(s_{t+1}) \right] \hat{\rho}_d^{ij}(t) + \mathbb{E}_{a_t \sim \pi_j}\left[ r(s_t, a_t) \right] - \gamma \mathbb{E}_{a_t \sim \pi_j} \mathbb{E}_{s_{t+1} \sim p_i}\left[ V^j(s_{t+1}) \right] = 0.$$

Hence, we can get the optimal dynamics importance weight, as follows:

$$\hat{\rho}_d^{ij}(t) = \left( \gamma V_j(s_{t+1}) - r_j(s_t, a_t) \right) \Big/ \left( 2\gamma \tilde{V}_{ij}^{DR}(s_{t+1}) \right).$$

### A.8. Proof of optimal dynamic weight that minimizing the variance

The variance can be rewritten as

$$Var_t\left[ \tilde{V}_{ij}^{DR}(s_t) \right] = \mathbb{E}_t\left[ \left( \rho_\pi^{ij}(t) \gamma \left( \hat{\rho}_d^{ij}(t) \tilde{V}_{ij}^{DR}(s_{t+1}) - \mathbb{E}_{t+1}[V^j(s_{t+1})] \right) - \rho_\pi^{ij}(t) \Delta(s_t, a_t) + \bar{V}_\theta(s_t, z_j) \right)^2 \right]$$
$$- \left( \mathbb{E}\left[ \tilde{V}_{ij}^{DR} \right] \right)^2 + \left( V(s_t) \right)^2 + \mathbb{V}(\rho_\pi), \tag{A.13}$$

where the second term is always negative and will be zero under $\hat{\rho}_d^{ij} = \frac{-r_j(s_t, a_t)}{\gamma \tilde{V}_{ij}^{DR}(s_{t+1})}$, which is nearly zero especially under the sparse-reward setting. Since the rest terms contain no $\hat{\rho}_d^{ij}(t)$, we consider the optimization of the first term in Eq. (A.13) w.r.t. $\hat{\rho}_d^{ij}(t)$, which is also the third term in Eq. (6) of Theorem 3.1 in the main text. We formulate the following optimization problem

$$\min_{\hat{\rho}_d^{ij}(t)} \quad \mathbb{E}_t\left[ \left( \rho_\pi^{ij}(t) \gamma \left( \hat{\rho}_d^{ij}(t) \tilde{V}_{ij}^{DR}(s_{t+1}) - \mathbb{E}_{t+1}[V^j(s_{t+1})] \right) - \rho_\pi^{ij}(t) \Delta(s_t, a_t) + \bar{V}_\theta(s_t, z_j) \right)^2 \right], \tag{A.14}$$

whose first-order derivative can be given as

$$2\mathbb{E}_t\left[ \gamma \rho_\pi^{ij}(t) \tilde{V}_{ij}^{DR}(s_{t+1}) \left( \rho_\pi^{ij}(t) \gamma \left( \hat{\rho}_d^{ij}(t) \tilde{V}_{ij}^{DR}(s_{t+1}) - \mathbb{E}_{t+1}[V^j(s_{t+1})] \right) - \rho_\pi^{ij}(t) \Delta(s_t, a_t) + \bar{V}_\theta(s_t, z_j) \right) \right] = 0.$$

Under the assumption that $\gamma \rho_\pi^{ij}(t) \tilde{V}_{ij}^{DR}(s_{t+1})$ is loosely correlated to the term in the parentheses, we have:

$$\mathbb{E}_t\left[ \left( \rho_\pi^{ij}(t) \gamma \left( \hat{\rho}_d^{ij}(t) \tilde{V}_{ij}^{DR}(s_{t+1}) - \mathbb{E}_{t+1}[V^j(s_{t+1})] \right) - \rho_\pi^{ij}(t) \Delta(s_t, a_t) + \bar{V}_\theta(s_t, z_j) \right) \right] = 0,$$

$$\mathbb{E}_t\left[ \rho_\pi^{ij}(t) \gamma \hat{\rho}_d^{ij}(t) \tilde{V}_{ij}^{DR}(s_{t+1}) - \rho_\pi^{ij}(t) \gamma \mathbb{E}_{t+1}[V^j(s_{t+1})] \right] + V_j(s_t) = 0,$$

$$\mathbb{E}_t\left[ \rho_\pi^{ij}(t) \gamma \hat{\rho}_d^{ij}(t) \tilde{V}_{ij}^{DR}(s_{t+1}) - \rho_\pi^{ij}(t) \gamma \mathbb{E}_{t+1}[V^j(s_{t+1})] + V_j(s_t) \right] = 0.$$

Hence, we can get the optimal importance weight as:

$$\hat{\rho}_d^{var}(t) = \big(\rho_\pi^{ij}(t)\gamma\mathbb{E}_{t+1}[V^j(s_{t+1})] - V^j(s_t)\big)/\big(\gamma\rho_\pi^{ij}(t)\tilde{V}_{ij}^{DR}(s_{t+1})\big)$$
$$= \big(\gamma\mathbb{E}_{t+1}[V^j(s_{t+1})] - V^j(s_t)/\rho_\pi^{ij}(t)\big)/\big(\gamma\tilde{V}_{ij}^{DR}(s_{t+1})\big).$$

We now review the variance in Eq. (A.13). When the increase of $\rho_\pi^{ij}(t)$ results in that $\hat{\rho}_d^{var}(t) > \frac{-r_j(s_t,a_t)}{\gamma\tilde{V}_{ij}^{DR}(s_{t+1})}$, continuously reducing $\hat{\rho}_d^{ij}(t)$ from $\hat{\rho}_\pi^{var}(t)$ to $\frac{-r_j(s_t,a_t)}{\gamma\tilde{V}_{ij}^{DR}(s_{t+1})}$ will enlarge the variance, since the first and the second terms in Eq. (A.13) will both increase. And reducing $\hat{\rho}_d^{ij}$ to near $\hat{\rho}_\pi^{var}(t)$ will reduce the variance.

**A.9. Proof of the upper bound for error of MSE**

Following the decomposition of MSE, we have

$$MSE(\tilde{V}^{DR}, \hat{\rho}_d^{ij*}) = Bias(\hat{\rho}_d^{ij*})^2 + Var_t(\tilde{V}^{DR}, \hat{\rho}_d^{ij*}),$$

$$MSE(\tilde{V}^{DR}, \hat{\rho}_d^{ij}) = Bias(\hat{\rho}_d^{ij})^2 + Var_t(\tilde{V}^{DR}, \hat{\rho}_d^{ij}).$$

Computing the bias of DR estimator using $\hat{\rho}_d^*$ and $\hat{\rho}_d^{var}$ separately and letting $\epsilon = \mathbb{E}_{a_t\sim\pi_j}\mathbb{E}_{s_{t+1}\sim p_i}\big[\gamma V^j(s_{t+1})\big] - \mathbb{E}_{a_t\sim\pi_j}\mathbb{E}_{s_{t+1}\sim p_j}\big[\gamma V^j(s_{t+1})\big]$, we have

$$Bias(\hat{\rho}_d^{ij*}) = \left|\mathbb{E}_{a_t\sim\pi_i}\mathbb{E}_{s_{t+1}\sim p_i}\big[\rho_\pi^{ij}(t)\big(\gamma V^j(s_{t+1}) - r_j(s_t,a_t)\big)/2 - \gamma\rho_\pi^{ij}(t)\rho_d^{ij}(t)V_{ij}^{DR}(s_{t+1})\big]\right|$$

$$= \left|\mathbb{E}_{a_t\sim\pi_j}\mathbb{E}_{s_{t+1}\sim p_i}\big[\gamma V^j(s_{t+1})\big]/2 - \mathbb{E}_{a_t\sim\pi_j}\big[r_j(s_t,a_t)\big]/2 - \mathbb{E}_{a_t\sim\pi_j}\mathbb{E}_{s_{t+1}\sim p_j}\big[\gamma V^j(s_{t+1})\big]\right|$$

$$= \left|\mathbb{E}_{a_t\sim\pi_j}\mathbb{E}_{s_{t+1}\sim p_i}\big[\gamma V^j(s_{t+1})\big]/2 + \mathbb{E}_{a_t\sim\pi_j}\big[r_j(s_t,a_t)\big]/2 - V^j(s_t)\right|$$

$$= \left|\mathbb{E}_{a_t\sim\pi_j}\mathbb{E}_{s_{t+1}\sim p_i}\big[\gamma V^j(s_{t+1})\big]/2 + \mathbb{E}_{a_t\sim\pi_j}\big[r_j(s_t,a_t)\big]/2 - V^j(s_t)\right|$$

$$= \left|\frac{\epsilon}{2} + \mathbb{E}_{a_t\sim\pi_j}\mathbb{E}_{s_{t+1}\sim p_j}\big[\gamma V^j(s_{t+1})\big]/2 + \mathbb{E}_{a_t\sim\pi_j}\big[r_j(s_t,a_t)\big]/2 - V^j(s_t)\right| = \left|\frac{\epsilon}{2} - \frac{V^j(s_t)}{2}\right|,$$

$$Bias(\hat{\rho}_d^{var}) = \left|\mathbb{E}_{a_t\sim\pi_i}\mathbb{E}_{s_{t+1}\sim p_i}\big[\rho_\pi^{ij}(t)\gamma\mathbb{E}_{t+1}[V^j(s_{t+1})] - V^j(s_t) - \gamma\rho_\pi^{ij}(t)\rho_d^{ij}(t)V_{ij}^{DR}(s_{t+1})\big]\right|$$

$$= \left|\mathbb{E}_{a_t\sim\pi_j}\mathbb{E}_{s_{t+1}\sim p_i}\big[\gamma V^j(s_{t+1})\big] - \mathbb{E}_{a_t\sim\pi_j}\mathbb{E}_{s_{t+1}\sim p_j}\big[\gamma V^j(s_{t+1})\big] - V^j(s_t)\right| = \left|\epsilon - V^j(s_t)\right|.$$

Hence, we have

$$\left|Bias(\hat{\rho}_d^{ij*})^2 - Bias(\hat{\rho}_d^{var})^2\right| = \frac{3}{4}\left|\epsilon - V^j(s_t)\right|^2, \quad \left|Bias(\hat{\rho}_d^{ij*})^2 - Bias(\rho_d^{ij})^2\right| = \frac{1}{4}\left|\epsilon - V^j(s_t)\right|^2,$$

and

$$\left|\epsilon - V^j(s_t)\right| = \left|\mathbb{E}_{a_t\sim\pi_j}\mathbb{E}_{s_{t+1}\sim p_i}\big[\gamma V^j(s_{t+1})\big] - \mathbb{E}_{a_t\sim\pi_j}\mathbb{E}_{s_{t+1}\sim p_j}\big[\gamma V^j(s_{t+1})\big] - V^j(s_t)\right|$$

$$= \left|\mathbb{E}_{a_t\sim\pi_j}\mathbb{E}_{s_{t+1}\sim p_j}\left[\left(\frac{1}{\rho_d^{ij}(t)} - 2\right)\gamma V^j(s_{t+1})\right] - \mathbb{E}_{a_t\sim\pi_j}\big[r_j(s_t,a_t)\big]\right|.$$

For the difference of variance, we have an upper bound as:

$$\left|Var_t(\tilde{V}^{DR}, \hat{\rho}_d^{ij*}) - Var_t(\tilde{V}^{DR}, \hat{\rho}_d^{ij})\right|$$

$$\leq \max\left(\left|Var_t(\tilde{V}^{DR}, \hat{\rho}_d^{ij*}) - Var_t(\tilde{V}^{DR}, \hat{\rho}_d^{var})\right|, \left|Var_t(\tilde{V}^{DR}, \hat{\rho}_d^{ij*}) - Var_t(\tilde{V}^{DR}, \rho_d^{ij})\right|\right)$$

$$= \max\left(\mathbb{E}_t\left[\big(A\hat{\rho}_d^{ij*}(t) - 2B\big)^2\right], \left|Var_t(\tilde{V}^{DR}, \hat{\rho}_d^{ij*}) - Var_t(V^{DR}, \rho_d^{ij})\right|\right).$$

*Table A.1.* Values set for the constant pair $(A, B)$ to generate random environment parameters.

| ALGORITHM | BODY MASS | BODY INERTIA | JOINT DAMPING | FRICTION |
|---|---|---|---|---|
| POINT-ROBOT-PARAMS-SPARSE | $(1.5, 1.0)$ | $(1.5, 1.0)$ | $(1.3, 1.0)$ | $(1.5, 1.0)$ |
| ANT-PARAMS-SPARSE | $(1.5, 3.0)$ | $(1.5, 3.0)$ | $(1.3, 3.0)$ | $(1.5, 3.0)$ |
| HUMANOID-PARAMS-SPARSE | $(1.5, 3.0)$ | $(1.5, 3.0)$ | $(1.3, 3.0)$ | $(1.5, 3.0)$ |
| HOPPER-PARAMS | $(1.5, 3.0)$ | $(1.5, 3.0)$ | $(1.3, 3.0)$ | $(1.5, 3.0)$ |
| WALKER-2D-PARAMS | $(1.5, 3.0)$ | $(1.5, 3.0)$ | $(1.3, 3.0)$ | $(1.5, 3.0)$ |
| POINT-ROBOT-PARAMS | $(1.5, 1.0)$ | $(1.5, 1.0)$ | $(1.3, 1.0)$ | $(1.5, 1.0)$ |
| SAWYER-PUSH-PARAMS-SPARSE | $(1.5, 2.5)$ | $(1.5, 2.5)$ | $(1.3, 2.5)$ | $(1.5, 2.5)$ |

Let $A = \gamma \rho_\pi^{ij}(t) \tilde{V}_{ij}^{DR}(s_{t+1})$ and $B = -\gamma \rho_\pi^{ij}(t) \mathbb{E}_{t+1}[V^j(s_{t+1})] - \rho_\pi^{ij}(t)\Delta(s_t, a_t) + V_\theta(s_t, z_j)$. Hence, we have the upper bound for the MSE difference between biased DR estimators using $\hat{\rho}_d^*$ and $\hat{\rho}_d$ as

$$\left| MSE(\tilde{V}_{ij}^{DR}, \hat{\rho}_d^{ij*}) - MSE(\tilde{V}_{ij}^{DR}, \hat{\rho}_d^{ij}) \right|$$

$$= \left| bias(\hat{\rho}_d^{ij*})^2 + Var_t(\tilde{V}^{DR}, \hat{\rho}_d^{ij*}) - bias(\hat{\rho}_d^{ij})^2 - Var_t(\tilde{V}^{DR}, \hat{\rho}_d^{ij}) \right|$$

$$\leq \left| bias(\hat{\rho}_d^{ij*})^2 - bias(\hat{\rho}_d^{ij})^2 \right| + \left| Var_t(\tilde{V}^{DR}, \hat{\rho}_d^{ij*}) - Var_t(\tilde{V}^{DR}, \hat{\rho}_d^{ij}) \right|$$

$$\leq \frac{3}{4} \left| \mathbb{E}_{a_t \sim \pi_j} \mathbb{E}_{s_{t+1} \sim p_j} \left[ \left( \frac{1}{\rho_d^{ij}(t)} - 2 \right) \gamma V^j(s_{t+1}) \right] - \mathbb{E}_{a_t \sim \pi_j} \left[ r_j(s_t, a_t) \right] \right|^2$$

$$+ \max \left( \mathbb{E}_t \left[ \left( A\hat{\rho}_d^{ij*}(t) - 2B \right)^2 \right], \left| Var_t(\tilde{V}^{DR}, \hat{\rho}_d^{ij*}) - Var_t(V^{DR}, \rho_d^{ij}) \right| \right).$$

# B. Experimental Setup

## B.1. Hyper-Parameters and Implementation Details

In our experiments, we utilize the same hyper-parameters for meta-training as in the open-sourced code of the baseline meta-RL approach, PEARL [4]. For our proposed DRT, we build up networks of predicting state transition for each task that has two layers with 500 units at each layer. The learning rate for the prediction network is $1e^{-3}$. We update the lower bound $\hat{\rho}_d^l$ at the end of each training epoch. Since negative transfer brought by reusing samples that may be inappropriately chosen by strategy $\mathcal{S}_I$ could result in a significantly lower target value as computed by DRaE for training, in practice, we take the maximum state value $\hat{V}(s)$ estimated by the value network $V_\theta$ and our DRaE $\tilde{V}^{DR}$ to further alleviate this issue.

## B.2. Generation of Varying Dynamics

The randomization of dynamics on all the environments in our experiments is implemented by generating different environment parameters through:

$$param_{ij} = \beta_j * init\_param_i, \tag{A.15}$$

where $\beta_j = A^{x_j}, x_j \sim Uniform(-B, B)$, $A$ and $B$ are the constants which control the generation of $\beta_j$ for each environment parameter of task $j$, and $init\_param_i$ is the initial value of the $i$-th environment parameter. Overall, these randomly sampled environment parameters include the body mass, body inertia, joint damping, and body component's friction, for which the values of constant pair $(A, B)$ are listed in Table A.1. With the initial values $init\_param_i$ loaded directly from the original file of "mujoco_py", the randomized environment parameter $param_{ij}$ is then obtained and set on the MuJoCo simulation engine to generate various environment dynamics.

## B.3. Reward Functions

For the dense-reward environments in Section 5.2, we use the same implementation as in PEARL's open-sourced code. For the sparse-reward environments with varying rewards and dynamics in Section 5.1, we modify their reward functions as:

$$reward = \begin{cases} -dist(robot, goal) + C & if \ dist(robot, goal) < D, \\ 0 & otherwise. \end{cases} \tag{A.16}$$

In the Point-Robot-Params-Sparse environment, we generate the goals in a cubic space, where we uniformly sample the goal coordinates in $(0.2, 0.5)$, $(−0.4, 0.4)$, and $(0.5, 1.5)$, and set $C = 1.0$ and $D = 0.2$. In the Ant-Params-Sparse environment, we uniformly generate the goals on a semi-circle with radius 2.0, and set $C = 4.0$ and $D = 0.8$. In the Humanoid-Params-Sparse environment, we uniformly generate the goals on a semi-circle with radius 3.0, and set $C = 3$ and $D = 0.8$. For all the environments, we keep the additional proprioceptive reward signal, which are control cost, contact cost, and survive bonus.