# OpenReview forum: "Doubly Robust Augmented Transfer for Meta-Reinforcement Learning"
_NeurIPS.cc/2023/Conference — NeurIPS 2023 poster_

### Official Review · Reviewer_gRwA · 2023-06-16

**Soundness:** 4 excellent
**Presentation:** 4 excellent
**Contribution:** 4 excellent
**Rating:** 7
**Confidence:** 3

**Summary:**

This study introduces the DRaT (Doubly Robust Augmented Transfer) algorithm, an advanced extension of hindsight-based transfer methods, which tackles not only reward mismatches but also discrepancies in transition dynamics. The authors provide a theoretical analysis to establish the optimality of their interval-based approximation, DRaE (Doubly Robust Augmented Estimator), in calculating the optimal importance sampling weight. Through empirical evaluation, it is demonstrated that DRaT surpasses conventional hindsight-based methods in performance, particularly in sparse-reward MuJoCo tasks that involve varying reward structures and transition dynamics.

**Strengths:**

**S1.** Addressing a Critical Yet Underexplored Aspect of Transfer Learning

This study tackles an essential but often overlooked aspect of transfer, specifically in the context of varying transition dynamics. The authors have put forth commendable efforts in highlighting the significance of calculating the optimal dynamics importance weight, which is a critical component in transfer under changing transition dynamics.

**S2.** Solid Theoretical Foundation and Novel Approximation Method

The work presents a robust theoretical underpinning for the proposed method, ensuring reliability and efficacy. The use of a tractable interval-based approximation is innovative and enhances the practicality of the approach. The proof confirming that the mean squared error (MSE) is guaranteed to encompass the optimum is both rigorous and convincing.

**S3.** Thorough Empirical Evaluation in Diverse Scenarios

The authors have conducted a comprehensive empirical evaluation by testing their method in three environments with sparse rewards and three with varying state dynamics. Such thorough testing provides a more complete picture of the algorithm's capabilities. Impressively, the proposed method demonstrates substantial and consistent improvements over all baselines, attesting to its effectiveness and robustness.

**Weaknesses:**

**W1.** Computational Overhead

A notable drawback of this study, which the authors have acknowledged in the limitations section, is the computational burden associated with calculating the doubly robust estimate and training the dynamics network. Given that practicality is an essential factor in the real-world application of algorithms, it is crucial to consider computational efficiency. It would be beneficial if the authors could include a comparison of the wall-clock time for each method, as this would provide valuable insights into the trade-offs between performance gains and computational costs.

**Acknowledgment Following Rebuttal**

I have reviewed the author's response. The response addressed my concerns.

**Questions:**

**Q1.** Results in Meta-RL are typically characterized by a high degree of variability, necessitating the aggregation of multiple runs for reliable evaluation. However, it appears that the number of seeds used for evaluation is not reported in the manuscript or the appendix. Could the authors clarify how many seeds were employed and consider including this information for a more transparent and rigorous evaluation?

**Q2.** Given the computational overhead associated with the proposed method, as discussed earlier, are there any strategies or optimizations that the authors have considered to enhance its computational efficiency? Insights into potential avenues for reducing computational costs without compromising effectiveness would be valuable.

**Q3.** Contrary to the checklist, I think the authors didn't submit the code or link to their codes. Any plans for releasing the code?

**Limitations:**

The authors have adequately addressed the limitations.

---

> ### Author Rebuttal · Authors · 2023-08-10
>
> **Comment 1 - Computational Overhead Analysis and Wall-Clock Time
> Comparison:**  Thanks for this valuable suggestion. Due to the space limit, please refer to our General Response to the Common Concern of "Providing Time Complexity Analysis of DRaT", for the detail discussion on the computational overhead analysis of our DRaT and wall-clock time comparison of all the methods.
>
> ****
>
> **Comment 2 - Strategies to Further Reduce Complexity of DRaT:** Thanks
> for this valuable suggestion. In the final version, we will provide some
> insights into potential avenues for reducing computational costs of our
> proposed DRaT, as follows.
>
> Here, we discuss about some possible strategies for further reducing the
> computational cost of DRaT without compromising too much of the
> effectiveness. For example, to reduce the computational cost of training
> dynamics prediction network for each training task, we may consider
> training a meta-dynamics prediction network, which can make prediction
> for all the tasks with a single network, and thus eliminate the need of
> training a separate dynamics prediction network for each task. To also
> reduce the computational cost of DR estimation, we may consider a
> similar solution in TD($n$), which makes a trade-off between TD($0$) and
> Monte-Carlo estimation by using the $n$-step rollout and fitted network.
>
>
> ****
>
> **Comment 3 - Choice of Random Seeds:** Thanks for this valuable
> suggestion. We will report in the final version that the results of each
> algorithm are averaged across 3 random seeds, which is also a common
> setting in the baseline meta-RL algorithms, such as PEARL \[4\] and
> PromP \[19\]. Please also note that the evaluation curves reported in
> this paper are obtained by evaluating on a test task set that contains
> dozens of tasks, which already reduces the randomness of evaluation in
> the meta-RL setting. To validate this, we further perform an experiment
> trial with 6 random seeds in the Ant-Params-Sparse and Humanoid-Params-Sparse environments. We
> then split the trail into two groups of 3 random seeds and average these
> two groups together. As shown in Fig. 4 of the additionally uploaded
> PDF, there is no significant difference of performance between these two
> groups.
>
>
> ****
>
> **Comment 4 - Plans for Releasing the Code:** We are now optimizing our
> source code, and will shortly release it right after we finish the code optimization.

---

> > ### Comment · Reviewer_gRwA · 2023-08-10
> > **Response to the Rebuttal**
> >
> > I'm grateful to the authors for addressing my concerns. Given the solid contribution of this work, I maintain my rating.

---

### Official Review · Reviewer_xMYP · 2023-07-03

**Soundness:** 3 good
**Presentation:** 4 excellent
**Contribution:** 4 excellent
**Rating:** 7
**Confidence:** 3

**Summary:**

The paper identifies that some existing approaches ignore  varying dynamics in meta-RL and proposes a new method for addressing this. In particular, the paper focuses on minimizing the mean-squared error of value functions and showed that doubly robust estimators suffer from a high-variance problem. Consequently, the paper proposes the doubly robust augmented estimator to estimate the importance weight of the dynamics, that reduces variance at the cost of introducing bias. The paper further identifies the intervals for which the importance weight should lie in, in order to maintain low variance. Finally, the paper provides empirical analysis to demonstrate that the proposed approach yields smaller value prediction error than existing baseline, and demonstrated that the proposed method can outperform existing baselines in certain control environments.

**Strengths:**

- The paper is well-written and easy to follow in general.
- Visualizations that supports various claims regarding the behaviours of the estimators---the paper demonstrate the proposed estimator has lower error compared to existing approach, and that the policy importance sampling weights do increase to approximately $1$ as informative trajectories are sampled more frequently.
- The paper demonstrated that the proposed method is theoretically sound (under some assumptions) and its empirical performance is comparable (if not better) to existing baselines.

**Weaknesses:**

## Major Comments
- Figure 3 suggests that with same reward function but different dynamics, the current baseline is already fairly robust? Does this really suggest that in figure 2, the problem is stemming from sparse reward?
	- This may have stemmed from a confusion. For section 5.2, page 9, line 329, the paper mentions that "..., while keeping all the other settings the same as in DRaT." Does that mean other approaches are now using DRaE?

**Questions:**

- The appendix mentions that the proofs really follow through by assuming that $\gamma \rho_{\pi}^{ij}(t) \tilde{V}_{ij}^{DR} (s)$ is not correlated to other expectation terms. I am wondering why this is a reasonable assumption---in particular, it is not immediately obvious that this term does not dominate the expectation terms.
- How does different $\sigma$ for approximating $N(s', \sigma)$ affect the estimation? Is there a sensitivity analysis regarding this? My expectation is that very small $\sigma$ would cause stability problem, while very high $\sigma$ would prevent learning. Is there an informed way to select this?


**Limitations:**

The paper has listed few limitations of the proposed approach.

---

> ### Author Rebuttal · Authors · 2023-08-10
>
> **Comment 1 - Concern on Figures 2 and 3:**
>
> Please note that though compared to Fig. 2, the current baselines (e.g.,
> PEARL \[4\] and HFR \[15\]) in Fig. 3 may have a smaller performance gap
> with our DRaT, they still suffer from a larger standard deviation of
> performance (as indicated by the larger shaded area in Figure 3). Such a
> larger standard deviation of performance would still affect their
> robustness. In comparison, our DRaT presents in general a smaller
> standard deviation of performance in Fig. 3.
>
> In comparison, the environments in Fig. 2 present an extremely
> challenging scenario with both the sparse-reward and varying dynamics
> settings. Specifically, the problem stemmed from sparse-reward is that
> the robot can only get reward signals indicating its distance to the
> goal position when it gets close to the goal position within a small
> range. On the other hand, the dynamics of this robot may also vary by
> setting different values of environment parameters, including the body
> mass, body inertia, joint damping, and body component's friction. As
> shown in Fig. 2, both the sparse-reward and varying dynamics settings
> bring challenges to the training of meta-RL algorithms.
>
> ****
>
> **Comment 2 - Confusion on Using DRaE in Other Approaches:** There might be
> some misunderstandings here. The original sentence in Line 329, Page 9
> is: "In IST, we use the IS estimator instead of our DRaE, while keeping
> all the other settings the same as in DRaT."
>
> Please note that this Importance Sampling augmented Transfer (IST)
> method is simply a variant of our Doubly Robust augmented Transfer
> (DRaT) method, with the same hyperparameter settings for network
> training and task relabeling as in DRaT, except for using the Importance
> Sampling (IS) estimator instead of the proposed Doubly Robust augmented
> Estimator (DRaE). While for the other comparison approaches, they still
> used their original value networks for value estimation, and did not
> utilize our proposed DRaE.
>
>
> ****
>
> **Comment 3 - Assumption on  $\gamma \rho^{ij}\_{\pi}(t) \tilde{V}^{DR}\_{ij}(s\_{t+1}) $ Loosely
> Correlated to Other Expectation Terms inside the Parentheses in Appendix
> A.7:** Thanks for pointing out this issue. In the following, we will
> briefly explain why this assumption is made, which will be further
> included in the final version.
>
> The original equation in Appendix A.7, after letting the first-order derivative of the objective function be zero, is given by:
>
> \begin{align*}
>     &2\mathbb E\_t \Bigg[ \Big(\gamma \rho^{ij}\_{\pi}(t) \Big)^2 \tilde{V}^{DR}\_{ij}(s\_{t+1}) \Big(\hat{\rho}^{ij}\_d(t) \tilde{V}^{DR}\_{ij}(s_{t+1}) - {\rho}^{ij}\_d(t) V^{DR}\_{ij}(s_{t+1}) \Big)\Bigg]
>     \\\\
> &+  2\mathbb{E}\_t \Bigg[ \gamma \rho^{ij}\_{\pi}(t)   \tilde{V}^{DR}\_{ij}(s_{t+1}) \Bigg( \rho^{ij}\_{\pi}(t)  \gamma \left( \hat{\rho}^{ij}\_d(t) \tilde{V}^{DR}\_{ij}(s\_{t+1}) -\mathbb{E}\_{t+1}[V^j(s\_{t+1})] \right)  - \rho^{ij}\_{\pi}(t)\Delta(s\_t,a\_t) +\bar{V}\_{\theta}(s\_t,z\_j)\Bigg) \Bigg] = 0.
> \end{align*}
> By eliminating the constant of 2 and further merging the two expectations on the left-hand side into one expectation, we have:
> \begin{align*}
>      \mathbb E\_t \Bigg[ \gamma \rho^{ij}\_{\pi}(t) \tilde{V}^{DR}\_{ij}(s\_{t+1}) & \cdot \Bigg(2\gamma \rho^{ij}\_{\pi}(t)\hat{\rho}^{ij}\_d(t) \tilde{V}^{DR}\_{ij}(s\_{t+1})
>     \\\\
>  &- \gamma \rho^{ij}\_{\pi}(t){\rho}^{ij}\_d(t) V^{DR}\_{ij}(s_{t+1}) -\gamma \rho^{ij}\_{\pi}(t)\mathbb{E}\_{t+1}[V^j(s\_{t+1})]  - \rho^{ij}\_{\pi}(t)\Delta(s\_t,a\_t) +\bar{V}\_{\theta}(s\_t,z\_j)\Bigg)\Bigg] =0.
> \end{align*}
> Note that inside expectation on the left-hand side is the multiplication of $\gamma \rho^{ij}\_{\pi}(t) \tilde{V}^{DR}\_{ij}(s\_{t+1}) $ with a summation enclosed by the parentheses, which can be rewritten as:
>
> \begin{align*}
> &\Bigg(2\gamma \rho^{ij}\_{\pi}(t)\hat{\rho}^{ij}\_d(t) \tilde{V}^{DR}_{ij}(s\_{t+1}) -\gamma \rho^{ij}\_{\pi}(t){\rho}^{ij}\_d(t) V^{DR}\_{ij}(s\_{t+1}) - \rho^{ij}\_{\pi}(t)\Delta(s\_t,a\_t) +\bar{V}\_{\theta}(s\_t,z\_j)\Bigg)
> \\\\
> =& \gamma \rho^{ij}\_{\pi}(t) \Bigg(2\hat{\rho}^{ij}\_d(t) \tilde{V}^{DR}\_{ij}(s\_{t+1}) - {\rho}^{ij}\_d(t) V^{DR}\_{ij}(s\_{t+1}) -\mathbb{E}\_{t+1}[V^j(s\_{t+1})] \Bigg)  - \rho^{ij}\_{\pi}(t)\Delta(s\_t,a\_t) +\bar{V}\_{\theta}(s\_t,z\_j).
> \end{align*}
>
> On the right-hand side of this equation, the first three terms are closely correlated to $\gamma \rho^{ij}\_{\pi}(t) \tilde{V}^{DR}\_{ij}(s\_{t+1}) $, while the last two terms are loosely correlated to it. Furthermore, given that $\hat{\rho}^{ij}\_d(t)$ is inside an interval upper-bounded by its true value ${\rho}^{ij}\_d(t)$, the value of $\hat{\rho}^{ij}\_d(t) \tilde{V}^{DR}\_{ij}(s_{t+1})$ is comparable to those of ${\rho}^{ij}\_d(t) V^{DR}\_{ij}(s\_{t+1})$ and $\mathbb{E}\_{t+1}[V^j(s\_{t+1})] $. Thus, the first three terms will be compensated by each other, while value of the last two terms $ - \rho^{ij}\_{\pi}(t)\Delta(s\_t,a\_t) +\bar{V}\_{\theta}(s\_t,z\_j)$ will dominate, which is loosely correlated to $\gamma \rho^{ij}\_{\pi}(t) \tilde{V}^{DR}\_{ij}(s\_{t+1}) $.
>
> ****
>
> **Comment 4 - Sensitivity Analysis of $\sigma$:** Thanks for this
> valuable suggestion. As will be included in the final version, we
> conduct experiments on the sensitivity analysis of $\sigma$ for the
> Ant-Params-Sparse environments. By varying the values of $\sigma$,
> we plot the curves of returns in Fig. 3 of the additionally
> uploaded PDF. As expected, a smaller value of $\sigma$ (e.g., when
> $\sigma=0.1$) would lead to a performance degradation at the initial
> training stage, but this performance gap will be reduced as the training
> proceeds. On the other hand, a larger value of $\sigma$ (e.g., when
> $\sigma > 2$) may incur a very high value for the value estimation and
> thus prevent the learning, the training curves of which are thus not
> reported in Fig. 3 of the additionally uploaded PDF.

---

> > ### Comment · Reviewer_xMYP · 2023-08-14
> >
> > I thank the authors for the detailed answers and have addressed my questions.

---

### Official Review · Reviewer_cabJ · 2023-07-04

**Soundness:** 3 good
**Presentation:** 3 good
**Contribution:** 3 good
**Rating:** 7
**Confidence:** 3

**Summary:**

This paper focuses on the meta-reinforcement learning setting with sparse reward. Previous work with hindsight-based sample transfer approaches requires the assumption that tasks differ only in reward functions. This paper proposes a doubly robust augmented transfer (DRaT) approach that allows both dynamics mismatches and varying reward functions across tasks. They first theoretically derive an upper bound of mean squared error between the estimated values of transferred samples and their true values in the target task. Then they propose an interval-based approximation to empirically find the optimal importance weight. In the experiment part, they implement DRaT on top of an off-policy meta-RL method and show that this method outperforms hindsight-based approaches on various sparse-reward MuJoCo locomotion tasks.

**Strengths:**

1.	This paper is well-written and easy to follow. The motivation for using doubly robust estimators to solve the sparse reward problem in meta-RL is convincing.
2.	The theoretical analysis of the doubly robust augmented estimator is detailed and well-motivated.
3.	The empirical implementation of the proposed algorithm outperforms other baselines by a large margin.


**Weaknesses:**

1.	The details about the meta-RL setting in experiments are not provided. The authors mention that they generate various dynamics by randomly sampling the environment parameters, including body mass, body inertia, joint damping, and body component friction. However, I cannot find details about the range of those parameters.
2.	The authors say that “One family contains the sparse-reward environments with varying reward functions and dynamics”. I cannot find details about how the reward functions are different in the settings.  Why does “control the arm of a 3D robot to reach random goals in the 3D space” has different reward functions?
3.	The algorithm requires the reward function for relabeling and selecting informative trajectories, which is similar to previous hindsight-based methods.


**Questions:**

1.	Please provide the missing details in the experiment as I mentioned in the weaknesses.
2.	Why use PEARL as the backbone? I am not familiar with the area of meta-RL with sparse reward, but I wonder if there is any later work that outputs PEARL in the meta-RL setting. If yes, maybe more baselines should be reported.


**Limitations:**

As discussed in the conclusion section, this method requires the reward function, which may not be available in some tasks. Since this is a common setting in existing methods for sparse reward meta-RL, it is still acceptable.

---

> ### Author Rebuttal · Authors · 2023-08-10
>
> **Comment 1- Providing Detailed Experiment Settings:**  Thanks for this
> valuable suggestion. We will provide in the final version more details
> about the meta-RL setting in experiments, as follows.
>
> The randomization of dynamics on all the environments in our experiments
> are implemented by generating different environment parameters through:
> $$
> param_{ij} = \beta_j*initparam_{i},
> $$
> where $\beta_j = A^{x_j}, x_j\sim Uniform(-B, B)$, $A$ and $B$ are the
> constants which control the generation of $\beta_j$ for each environment
> parameter of task $j$, and $initparam_{i}$ is the initial value of the
> $i$-th environment parameter. Overall, these randomly sampled
> environment parameters include the body mass, body inertia, joint
> damping, and body component's friction, for which the values of constant
> pair $(A,B)$ are listed in Table 2 of the additionally uploaded PDF.
> With the initial values $initparam_{i}$ loaded directly from the
> original file of "mujoco_py", the randomized environment parameter
> $param_{ij}$ is then obtained and set on the MuJoCo simulation engine to
> generate various environment dynamics.
>
> Please note that since the robot models in our testing environments are
> complicated with a large number of environment parameters, these initial
> values of $initparam_{i}$ are usually stored as high-dimensional
> arrays in the "mujoco_py" file. Due to space limit, we only show the
> initial values set for the Ant-Params-Sparse environments in Fig. 2 of
> the additionally uploaded PDF as an example. For more details about
> settings of these initial values, please refer to either the "mujoco_py"
> file that is publicly available, or our source code that will be
> released shortly after the code optimization.
>
> ****
>
> **Comment 2 - Using PEARL as Backbone:** We adopted PEARL as the off-policy
> meta-RL backbone in this paper, mainly because it is a sample-efficient
> meta-RL algorithm, which significantly outperforms the other baseline
> meta-RL algorithms, such as MAML \[12\], PromP \[19\] and RL2 \[3\] on
> the standard meta-RL testing benchmark. This also explains why the
> following works that further tackle the sparse-reward challenge in
> meta-RL are mostly built upon the PEARL backbone, such as the Hindsight
> Task Relabeling (HTR) \[14\] and Hindsight Foresight Relabeling (HFR)
> \[15\]. These methods have demonstrated a superiority over PEARL in the
> sparse-reward setting, and also were compared with our DRaT in the
> experiments section.
>
>
> [3] Y. Duan, J. Schulman, X. Chen, P. L. Bartlett, I. Sutskever, and P. Abbeel. RL2: Fast reinforcement learning via slow reinforcement learning. arXiv preprint arXiv:1611.02779, 2016.
>
> [12] C. Finn, P. Abbeel, and S. Levine. Model-agnostic meta-learning for fast adaptation of deep networks. In ICML, 2017.
>
> [14] C. Packer, P. Abbeel, and J.h E Gonzalez. Hindsight task relabelling: Experience replay for sparse reward meta-RL. In NeurIPS, 2021.
>
> [15] M. Wan, J. Peng, and T. Gangwani. Hindsight foresight relabeling for meta-reinforcement learning. In ICLR, 2022.
>
> [19] J. Rothfuss, D. Lee, I. Clavera, T. Asfour, and P. Abbeel. Promp: Proximal meta-policy search. In ICLR, 2019.
>
> ****
>
> **Comment 3 - Details about Settings of Different Reward Functions:** Thanks
> for pointing out this issue. In Section B.2 of the Supplementary
> Material, we had provided the reward function settings for the
> sparse-reward environments with varying rewards and dynamics, while the
> reward function settings for the dense-reward environments were stated
> as using the same implementation as in PEARL's open-sourced code. In the
> following, we will provide further explanation to make the settings of
> different reward functions clearer.
>
> The reward function defined for the Point-Robot environments, which
> control the arm of a 3D robot to reach a random goal position in the 3D
> space, is given as: $$reward=
> 		\begin{cases}
> 			-dist(robot,goal)+C  &     & if\ dist(robot,goal)<D,\\\\
> 	     0  &     & otherwise.\\\\
> 		\end{cases} $$ We set $C=0.0$, $D=+\infty$ for the dense-reward
> function in Point-Robot-Params, where the reward signal stands for the
> distance between the tip of robot arm and the goal position, which can
> always be accessed. We set $C=1.0$ and $D=0.2$ for the sparse-reward
> function in Point-Robot-Params-Sparse, where the RL agent will get a
> meaningful reward signal that indicates the distance to the goal only
> when the robot arm is close to its goal within a small range. In both
> cases, goal positions are randomly generated in the 3D space, and hence
> the reward functions are also varying to indicate a distinct distance to
> different goal positions.
>
> ****
>
> **Comment 4 - Concern on Requiring Reward Function for Relabeling and
> Selecting Informative Trajectories:** Please note that the commonly used
> hindsight-based methods, such as HTR \[14\] and AIR \[6\], simply use
> the reward function to relabel samples without accommodating to the
> dynamics mismatch across tasks. Our experiments in Section 5.1
> demonstrated such an inefficiency of HTR and AIR on the extremely
> challenging environments with both sparse-reward and varying dynamics
> settings.
>
> Different from them, we design a doubly robust augmented estimator
> (DRaE) to accommodate to the more general and rational meta-RL setting
> with both sparse-reward and varying dynamics across different tasks.
> DRaE can tackle the mismatch of dynamics distributions in meta-RL with a
> guaranteed optimum for the dynamics importance weight by minimizing MSE
> between the estimated and true values of the value function. Using DRaE
> for a better value estimation of transferred samples, our proposed DRaT
> algorithm demonstrated its superiority on several meta-RL testing
> benchmarks, as verified by the experiments in Sections 5.1 and 5.2.

---

> > ### Comment · Reviewer_cabJ · 2023-08-10
> > **Response to authors**
> >
> > Thanks for addressing my concerns. The experiment setting is now clear to me. I don't have further questions and I am happy to increase my score to a clear accept.

---

### Official Review · Reviewer_L57L · 2023-07-04

**Soundness:** 3 good
**Presentation:** 3 good
**Contribution:** 3 good
**Rating:** 6
**Confidence:** 3

**Summary:**

The paper introduces Doubly Robust Augmented Transfer (DRaT), a novel approach for dealing with sparse-reward scenarios in meta-reinforcement learning (Meta-RL). DRaT transfers informative trajectories from various tasks to a target task, effectively handling dynamics mismatches and different reward functions. The authors propose an interval-based approximation to the importance weight. The DRaT algorithm is applied to an off-policy Meta-RL baseline, demonstrating superior performance over other hindsight-based methods on various sparse-reward MuJoCo tasks with different dynamics and reward functions.

**Strengths:**

**Originality**: The paper presents a novel approach, Doubly Robust Augmented Transfer (DRaT), for dealing with sparse-reward scenarios in meta-reinforcement learning. The idea of transferring informative trajectories from various tasks to a target task is an interesting direction in the field of meta-reinforcement learning.

**Quality**: DRaT effectively handles dynamics mismatches and different reward functions across tasks, for the MuJoCo environments it is tested on. The method outperforms other hindsight-based methods on various sparse-reward tasks. However, without a detailed analysis of the time complexity of the algorithm or a comparison with other state-of-the-art methods in terms of computational efficiency, it's hard to fully assess the quality of the method.

**Clarity**: While the paper presents complex ideas, it does so in a clear and understandable manner. The authors have done a good job of explaining their method and its benefits. However, some sections could benefit from more detailed explanations or examples, particularly for readers who are not deeply familiar with the field.

**Significance**: The significance of the paper is evident in its potential impact on the field of meta-reinforcement learning. DRaT addresses a key challenge in the field - dealing with sparse-reward scenarios. By demonstrating superior performance on various tasks, the paper shows that the method could be a valuable tool for researchers and practitioners in the field. However, the real-world applicability and scalability of the method would need to be further explored to fully understand its significance.

**Weaknesses:**

**More Extensive Evaluation**: While the paper demonstrates the effectiveness of DRaT on various sparse-reward tasks, a more extensive evaluation would strengthen the paper's claims. The current evaluation is somewhat limited and does not fully explore the method's performance across a wide range of scenarios. The authors could consider using the [MetaWorld](https://arxiv.org/pdf/1910.10897.pdf) benchmark, a comprehensive benchmark of environments specifically designed for meta-reinforcement learning. This would provide a more rigorous testing ground for the DRaT method.

**Time Complexity Analysis**: The paper does not provide detailed information on the time complexity of DRaT. Information regarding the performance with respect to time in the experiments would be valuable in assessing the method's scalability and efficiency, particularly for large-scale problems.

**Questions:**

1. Is a separate dynamics prediction network fitted for each possible task 'j' in the training set?
2. Would it be possible to test the DRaT method using the MetaWorld benchmark, which is specifically designed for meta-reinforcement learning?
3. Could you provide a detailed analysis of the time complexity of the DRaT method to assess its scalability and efficiency?

**Limitations:**

The authors adequately addressed the limitations.

---

> ### Author Rebuttal · Authors · 2023-08-10
>
> **Comment 1 - Providing Time Complexity Analysis of DRaT:** Thanks for this valuable suggestion. Due to the space limit, please refer to our General Response to the Common Concern of "Providing Time Complexity Analysis of DRaT" for the detail discussion on the time complexity analysis of DRaT, wall-clock time comparison of all the methods, and possible strategies for further reducing the computational cost of DRaT .
>
> ****
>
> **Comment 2 - More Evaluation on MetaWorld Benchmark:** We reported in this paper an extensive
> evaluation of our DRaT on MuJoCo environments, which are the commonly
> adopted benchmark to evaluate meta-RL, such as in the meta-RL baselines
> like PEARL \[4\], MAML \[12\], and PromP \[19\]. For a fair comparison
> with PEARL, and with other comparison methods that are built upon PEARL
> (e.g., HTR, HFR, AIR), we thus also adopt MuJoCo as the benchmark for
> empirical evaluations. Please also note that the testing environments
> adopted in Section 5.1 are extremely challenging with both the
> sparse-reward and varying dynamics settings, where the baseline (PEARL)
> or comparison algorithm (HTR) may even fail to learn a useful policy.
> While our DRaT presents a significant performance gain, demonstrating
> its effectiveness in these challenging environments.
>
> Following the reviewer's advice, we have also tried to test our DRaT
> using the MetaWorld benchmark. Due to the time constraint, we managed to
> conduct experiments on a pushing task in MetaWorld, which controls a
> simulated Sawyer arm to push a block to a specified goal position. We
> also modify the original sparse-reward Sawyer-Push to
> Sawyer-Push-Params-Sparse by generating different dynamics for each
> sampled task. We compare DRaT with HFR and PEARL in Fig. 1 of the
> additionally uploaded PDF, which is because they show better performance
> than other baselines in MuJoCo, and also because HFR is the only one of
> all the comparison methods that is tested in MetaWorld. Our DRaT still
> presents a better performance than HFR and PEARL, while PEARL even fails
> to improve its policy.
>
> [4] K. Rakelly, A. Zhou, C. Finn, S. Levine, and D. Quillen. Efficient off-policy meta-reinforcement learning via probabilistic context variables. In ICML, 2019.
>
> [12] C. Finn, P. Abbeel, and S. Levine. Model-agnostic meta-learning for fast adaptation of deep networks. In ICML, 2017.
>
> [19] J. Rothfuss, D. Lee, I. Clavera, T. Asfour, and P. Abbeel. Promp: Proximal meta-policy search. In ICLR, 2019.
>
> ****
>
> **Comment 3 - Concern on Dynamics Prediction Network:** As in Line 2 of our
> proposed DRaT in Algorithm 1, a separate dynamics prediction network needs to
> be fitted for each sampled training task. Please refer to our General Response to the Common Concern
> of "Providing Time Complexity Analysis of DRaT" for the additional complexity
> introduced by doing so, where we also discussed about possible strategy
> that can be used to reduce this part of complexity, i.e., by training a
> meta-dynamics prediction network to make prediction for all the tasks
> with a network, and thus eliminating the need of training a separate
> dynamics prediction network for each task.

---

> > ### Author Response · Authors · 2023-08-15
> >
> > Dear Reviewer,
> >
> >
> > We appreciate your valuable suggestions which helped us in improving our paper. Since the end of discussion period is approaching, we were just wondering if there is any further question that we would like to answer at any time.
> >
> >
> > Thanks again for the time and effort you have dedicated to reviewing our paper and providing these insightful comments.

---

> > > ### Comment · Reviewer_L57L · 2023-08-19
> > >
> > > Thank you for the detailed response. I consider you have addressed my concerns, so I have increased my score accordingly.

---

### Author Rebuttal · Authors · 2023-08-10

**General Response:** We would like to thank all the reviewers for their helpful comments. Here, we will respond to the common concern on the time complexity of our proposed DRaT. For orther concerns, please see below our responses to each reviewer’s individual comments, where the newly added figures and tables can be found in our additionally uploaded PDF attached at the end of this General Response.

****

**Common Concern - Providing Time Complexity Analysis of DRaT:** As had been acknowledged in the limitations
section, additional computational complexity would be brought by our
DRaT for computation of DR estimate and training of dynamics prediction
network for each task. Here, **1)** we analytically show that this
additional complexity is comparable to its baseline PEARL's
computational complexity, with a linear scaling factor of
$\frac{\max(N_{\mathcal{B}}, L) }{N_{\mathcal{B}}} \cdot \frac{\max(c_1, c_2, c_3)}{c_1}$.
**2)** We then empirically verify this analysis by demonstrating that
the wall-clock time of DRaT is $1.3\times \sim 1.7\times$ that of PEARL,
and compare the wall-clock time of other state-of-the-art methods.
**3)** We also discuss about some possible strategies for further
enhancing DRaT's computational efficiency. These analyses will be
included in the final version.


**1) Complexity Analysis.**


Typically in an off-policy meta-RL algorithm, each epoch (i.e., meta-training iteration) is divided into the sampling and training phases. In the sampling phase, the computational cost stems mainly from the actions chosen by feed-forward computation of policy network and inference network. Since all the algorithms (i.e., HTR, HFR, AIR, DRaT) follow the same sampling process as PEARL, we only analyze the computational cost in the training phase.

**PEARL's complexity:** In the training phase, the computational cost contains mainly the feed-forward and back-propagation computation of policy network, value network and inference network. Given $K$ training iterations at each epoch, batch size $N_{\mathcal{B}}$ of transitions from $N$ training tasks, state space cardinality $\vert \mathcal{S} \vert$ and action space cardinality $\vert \mathcal{A} \vert$, and assuming a constant computational cost of feed-forward and back-propagation computation $c_1$, the total computational cost of training phase is $O(K \cdot N_{\mathcal{B}} \cdot N \cdot \vert \mathcal{S} \vert \cdot \vert \mathcal{A} \vert \cdot c_1 )$.

**DRaT's complexity:** Besides the same training cost $O(K \cdot N_{\mathcal{B}} \cdot N \cdot \vert \mathcal{S} \vert \cdot \vert \mathcal{A} \vert \cdot c_1 )$ as PEARL, additional computational cost brought by DRaT includes training of dynamics prediction networks, computation of DR estimator, and computation of relabelling.

- Considering building separate prediction networks for $N$ training tasks and the cost of feed-forward and back-propagation computation $c_1$, using the same batch of transitions for training, the additional cost of training dynamics prediction networks is also $O(K \cdot N_{\mathcal{B}} \cdot N \cdot \vert \mathcal{S} \vert \cdot \vert \mathcal{A} \vert \cdot c_1 )$.

- Considering sampling informative trajectories with a maximal length of $L$ for $N$ training tasks and the cost $c_2$ of DR estimation computation at each time step, the computational cost of DR estimation is $O(K \cdot N \cdot L \cdot  \vert \mathcal{S} \vert \cdot \vert \mathcal{A} \vert \cdot c_2)$.

- In this paper, we used the approximate inverse RL relabeling (AIR). We sample one trajectory from each training task for relabeling, leading to $N$ candidate trajectories in total. Assuming the cost of computing relabeled reward at each time step is $c_3$, the computational cost of relabeling is $O(K \cdot N \cdot L \cdot \vert \mathcal{S} \vert \cdot \vert \mathcal{A} \vert \cdot c_3)$.


Incorporating them together, we conclude that the additional computational complexity of DRaT is dominated by $O\Big(K \cdot N \cdot  \max ( N_{\mathcal{B}}, L ) \cdot \vert \mathcal{S} \vert \cdot \vert \mathcal{A} \vert \cdot \max ( c_1, c_2, c_3 ) \Big)$, which is comparable to PEARL with a linear scaling factor of $\frac{\max(N_{\mathcal{B}}, L) }{N_{\mathcal{B}}} \cdot \frac{\max(c_1, c_2, c_3)}{c_1}$.

**2\) Wall-Clock Time Comparison.**

To empirically verify the above complexity analysis, in Table 1 of the
additionally uploaded PDF, we also compare the wall-clock time for DRaT
and all the comparison methods, including PEARL, HTR, HFR, AIR. It can
be seen that for these robotic control tasks at different scales, DRaT
generally consumes $1.3\times \sim 1.7\times$ running time of PEARL,
which is also comparable to other baselines.

**3\) Strategies to Further Reduce Complexity.**

Here, we discuss about some possible strategies for further reducing the
computational cost of DRaT without compromising too much of the
effectiveness. For example, to reduce the computational cost of training
dynamics prediction network for each training task, we may consider
training a meta-dynamics prediction network, which can make prediction
for all the tasks with a single network, and thus eliminate the need of
training a separate dynamics prediction network for each task. To also
reduce the computational cost of DR estimation, we may consider a
similar solution in TD($n$), which makes a trade-off between TD($0$) and
Monte-Carlo estimation by using the $n$-step rollout and fitted network.

****

---

### Decision · Program_Chairs · 2023-09-21

**Decision:**

Accept (poster)

**Comment:**

The review scores are 7, 7, 7, 6, with an average score of 6.75. The reviewers’ comments are generally positive. The authors have given a detailed response to the questions and comments, and the reviewers have positively responded to the comments.

For the final version, please make sure that you include the comments and suggestion by the reviewers.

Regards,

AC